# Optofluidic three-dimensional microfabrication and nanofabrication

Three-dimensional (3D) microfabrication/nanofabrication technologies have attracted tremendous attention as they can create various functional microdevices/nanodevices, such as microrobots[1,2], micro-actuators[3], micrometre-scale metamaterials[4,5,8] and microphotonic/nanophotonic devices[6]. Over the past two decades, two-photon polymerization (2PP) has emerged as the state-of-the-art 3D microfabrication/nanofabrication strategy owing to its high resolution (up to 100 nm), facile fabrication process and capability of printing intricate free-form 3D microstructures/nanostructures[9–12]. However, the material compatibility of 2PP remains highly limited to cross-linkable polymers. Recent efforts have aimed to expand printable materials beyond polymers, primarily through developing advanced photoresists with tailored chemistries, such as grafting photochemically bonding ligands onto inorganic colloidal nanocrystals[13,14] or incorporating metal-coordination complexes into cross-linkable monomers[15–21]. Nonetheless, these approaches remain restricted to specific materials and continue to face challenges in achieving broad compatibility with diverse material systems.

As a promising approach to overcome these material limitations, direct assembly of material building blocks has proved effective for 3D microstructures/nanostructures[22]. Among various assembly techniques, optical assembly[23–30] has been an appealing strategy for the construction of complex material assemblies. It uses the non-specific light–matter interactions (for example, optical gradient forces[31] and light-controlled electric or temperature fields[32]) to trap microparticles/nanoparticles suspended in a solution. The trapped particles can then be transported and positioned one by one at designated locations, enabling high-precision single-particle assembly. Techniques such as optical tweezers[33], opto-thermophoretic assembly[34], opto-thermoelectric assembly[35] and optothermal-flow-based assembly[24,27,36,37] have been used to assemble colloidal particles with diverse sizes, shapes and surface properties. However, these approaches remain largely limited to 2D structural configurations and exhibit low assembly efficiencies, typically on the order of $10^1$–$10^3$ particles $min^{-1}$. More importantly, establishing a general optical assembly platform with broader material applicability is still challenging, as most techniques impose strict requirements on parameters, such as particle size/surface properties, solvent composition and other environmental conditions. A detailed comparison of various assembly strategies is provided in Supplementary Table 1. Therefore, innovations

**Fig. 1 | Concept of optofluidic 3D microfabrication/nanofabrication. a**, Schematic illustration of the optofluidic 3D microfabrication/nanofabrication process, in which a localized temperature gradient induced by femtosecond (fs) laser heating generates a strong convective flow, guiding the 3D assembly of microparticles/nanoparticles within a confined hollow 3D microtemplate, printed by 2PP. **b,c**, SEM images of a $SiO_2$ colloidal-particle-assembled microcube (**b**) and its zoomed-in view (**c**). **d,e**, SEM images of a dangling croissant-shaped microstructure with 3D curved surface assembled from $SiO_2$ particles (**d**) and its enlarged view (**e**). The inset in **d** is the 3D model of the croissant structure. **f**, Simulation result showing the temperature distribution and fluid flow field around a hollow microcube following fs laser heating. **g**, Schematic illustrations and time-lapse optical images showing the assembly process of $SiO_2$ nanoparticles within a hollow microcube. These images are extracted from Supplementary Video 1. Scale bars, 10 μm (**b**,**d**); 4 μm (**c**,**e**); 20 μm (**g**).

in general assembly mechanisms are urgently needed to achieve controllable, non-specific guidance and precise assembly of material building blocks at the microscale/nanoscale, enabling the creation of high-quality, truly volumetric free-form 3D architectures with broad material compatibility.

Here we propose a universal 3D microfabrication/nanofabrication strategy compatible with a broad range of materials. This technique uses the optofluidic interaction—light-driven flow—to efficiently assemble diverse micromaterials/nanomaterials into predefined 2PP-printed microtemplates, creating high-quality, truly volumetric free-form 3D microarchitectures/nanoarchitectures from a variety of materials, including diamond, metal, metal oxide, quantum dots and others. By integrating the optofluidic assembly process with 2PP, our optofluidic 3D microfabrication/nanofabrication strategy proposes a generalizable model that overcomes the material limitations of 2PP,

opening new avenues for nanotechnology (Supplementary Fig. 1 and Supplementary Table 2).

## Concept of optofluidic 3D micro-/nanofabrication

The working process of optofluidic 3D microfabrication/nanofabrication is illustrated in Fig. 1a and consists of the following key steps. First, a 3D hollow polymeric microstructure with an open hole (for example, a cube) is printed on a glass substrate by the 2PP process, serving as the 3D microtemplate. Next, the printed template is immersed in a solution containing uniformly dispersed nanoparticles (or few-micrometre particles). A femtosecond (fs) laser with a beam diameter of 2 μm is then applied near the open hole of the template, generating a sharp temperature gradient that induces a strong convective flow (up to several mm s⁻¹), propelling the dispersed particles towards the open hole. As a

result, these particles are transported into the hollow microtemplate and accumulate over time, ultimately assembling into the prescribed 3D shape dictated by the template. Following assembly, the polymer template is selectively removed through rational post-treatments (see details in Methods), yielding a free-standing 3D volumetric microarchitecture composed entirely of the targeted materials. For example, our approach enables the fabrication of a series of solid microcubes randomly assembled by $SiO_2$ nanoparticles with an assembly efficiency of approximately $10^5$ particles min[−1], as shown in the scanning electron microscopy (SEM) image in Fig. 1b, its enlarged image in Fig. 1c and Supplementary Fig. 2. The resulting 3D microarchitecture exhibits high structural integrity, as these constituent nanoparticles are strongly bonded and stabilized by van der Waals forces. Even without further chemical bonding or annealing process for interfacial improvement, these 3D structures can be self-supporting and mechanically stable, owing to the strong van der Waals interactions in colloidal nanoparticle assemblies[12]. This is further evidenced by the successful fabrication of a dangling croissant-shaped superstructure with intricate 3D curved surfaces (Fig. 1d,e), demonstrating the robustness of this technique.

To gain deeper insights into the assembly process, we conducted simulations and recorded the experimental process to analyse the underlying dynamics. As shown in Fig. 1f, Supplementary Fig. 3 and Supplementary Video 1, when laser heating is applied at the open hole of a hollow microcube, a sharp thermal gradient near the hotspot is induced. This thermal gradient causes variations in fluid density, pressure and surface tension around the hole[26,38], generating a strong, directed fluid flow towards the open hole, transporting dispersed $SiO_2$ particles into the microcube. Furthermore, we note that bubble formation with a diameter up to 100 μm (Supplementary Fig. 4) occurs easily owing to solvent evaporation induced by femtosecond laser heating, leading to an extra flow from the Marangoni effect[39], and this disturbance further promotes and accelerates particle assembly within the template (Supplementary Video 1). As shown in Fig. 1g and Supplementary Video 1, an assembly speed of around 700 μm³ s[−1] is reached when assembling $SiO_2$ colloidal particles with a diameter of 1 μm, which is about twice as fast as the typical 2PP process (Supplementary Fig. 5). By precisely manipulating the light-driven fluid flow through localized laser irradiation, any nanoparticles carried in the flow can be efficiently assembled into 3D microarchitectures. This non-selective mechanism makes the strategy inherently adaptable to a broad range of materials.

## Assembly mechanism

The assembly process is primarily governed by two competing physical forces: inter-particle interactions and particle–fluid interactions. We start with a model system in which $SiO_2$ colloidal particles (150 nm in diameter) assemble in aqueous solutions with varying ionic strengths. In this system, the inter-particle interactions consist of van der Waals attraction and electric double layer (EDL) repulsion, commonly described as the net DLVO interaction force[40,41]. The inter-particle DLVO force is highly sensitive to particle distance and particle properties and plays a crucial role in colloidal cluster formation. Meanwhile, the particle–fluid interactions are represented by hydrodynamic forces, particularly the Stokes drag force, which reflects how particles are carried by the surrounding flow. In our local assembly region, it tends to disperse the particles and counteracts aggregation (Fig. 2a). To determine whether colloidal particles will form clusters, we analyse the balance between inter-particle potential energy and hydrodynamic effects. The total interaction energy governing the system can be expressed as: $U_{total} = U_{DLVO} + W_{fluid}$, in which $U_{DLVO}$ represents the interaction potential between particles and $W_{fluid}$ denotes the work done by hydrodynamic forces. The DLVO potential energy is derived from the integration of the inter-particle DLVO force over the separation distance, whereas the work done by the fluid force depends on the external flow field and particle motion. The tendency of the system to form clusters can then be evaluated by examining changes in $U_{total}$. If $\Delta U_{total} < 0$, the inter-particle attractive interactions dominate, leading to energetically favourable conditions for colloidal clustering around the laser spot. If $\Delta U_{total} > 0$, particle–fluid interactions prevail, causing particles to move with the fluid flow and remain dispersed (Fig. 2a).

The template we used plays a crucial role in enabling deterministic 3D fabrication: it defines the overall geometry to ensure well-defined edges and symmetry, channels the optofluidic flow to reproducibly fill complex volumes and provides design versatility by allowing diverse 3D architectures. To effectively assemble dispersed $SiO_2$ colloidal particles within a confined 3D space, the system must satisfy the condition $\Delta U_{total} < 0$ to promote colloidal cluster formation. This can be achieved through two approaches: (1) increasing the inter-particle attractive interaction or (2) decreasing the speed of fluid flow. We first investigate the effect of varying inter-particle interactions on $SiO_2$ cluster formation. In aqueous solution, $SiO_2$ particles are highly surface charged, resulting in strong electrostatic repulsion that stabilizes the dispersion[42–44]. By tuning the ionic strength of the solution, the zeta potential of $SiO_2$ colloidal particles and the Debye length—a characteristic length scale over which electric fields are naturally screened—can be systematically adjusted, thereby influencing inter-particle electrostatic interactions. As illustrated in Fig. 2b, increasing the NaCl concentration progressively decreases the zeta potential of $SiO_2$ particles and compresses the Debye length, thus reducing the electrostatic repulsion within the EDL. Consequently, the attractive component of the DLVO interaction is enhanced, leading to stronger particle aggregation. At higher NaCl concentrations, this effect facilitates the formation of larger $SiO_2$ clusters, whereas no notable clustering is observed when the NaCl concentration is less than 200 mM. Correspondingly, as depicted in Fig. 2c, $SiO_2$ particles effectively assemble and accumulate within a hollow template in a solution with 1 M NaCl concentration (Supplementary Video 2, part 1), whereas no assembly is observed at a lower NaCl concentration (100 mM; Fig. 2d and Supplementary Video 2, part 2).

Flow speed is another vital factor affecting the cluster formation of the $SiO_2$ colloidal particles. Higher flow speeds generate stronger Stokes forces acting on colloidal particles, hindering their aggregation. By comparing the relative values between $U_{DLVO}$ and $W_{fluid}$, a critical flow speed of approximately 300 μm s[−1] is theoretically determined and summarized in a phase diagram (Fig. 2e; see Methods for details). Above this critical speed, the Stokes force dominates and completely overcomes the DLVO attraction, preventing cluster formation regardless of NaCl concentration. Below this threshold, the DLVO force becomes comparable to the Stokes force, allowing cluster formation at high NaCl concentrations (for example, >0.5 M). By adjusting the laser scan speed, different flow speeds can be generated experimentally. Overall, the assembly process was systematically conducted and analysed across various NaCl concentrations and flow speeds. The experimental results (represented as discrete symbols in Fig. 2e) closely align with theoretical predictions. For example, $SiO_2$ particles in 1 M NaCl solution aggregate into clusters at flow speeds less than 320 μm s[−1], whereas they remain dispersed at higher flow speeds of about 900 μm s[−1] (Supplementary Fig. 6 and Supplementary Video 3). Besides, an extremely weak flow field cannot provide sufficient driving force to continuously transport the particles towards the inside of a template, failing to fill up the template (Supplementary Video 4).

## Assembly in various solvent systems

To broaden the compatibility of the optofluidic 3D microfabrication/nanofabrication technique, the 3D assembly of microparticles/nanoparticles can be extended to various solvent systems, as different colloidal particles in different solutions exhibit varying balances between inter-particle and particle–fluid interactions. For instance, replacing a good solvent with a poor solvent can enhance

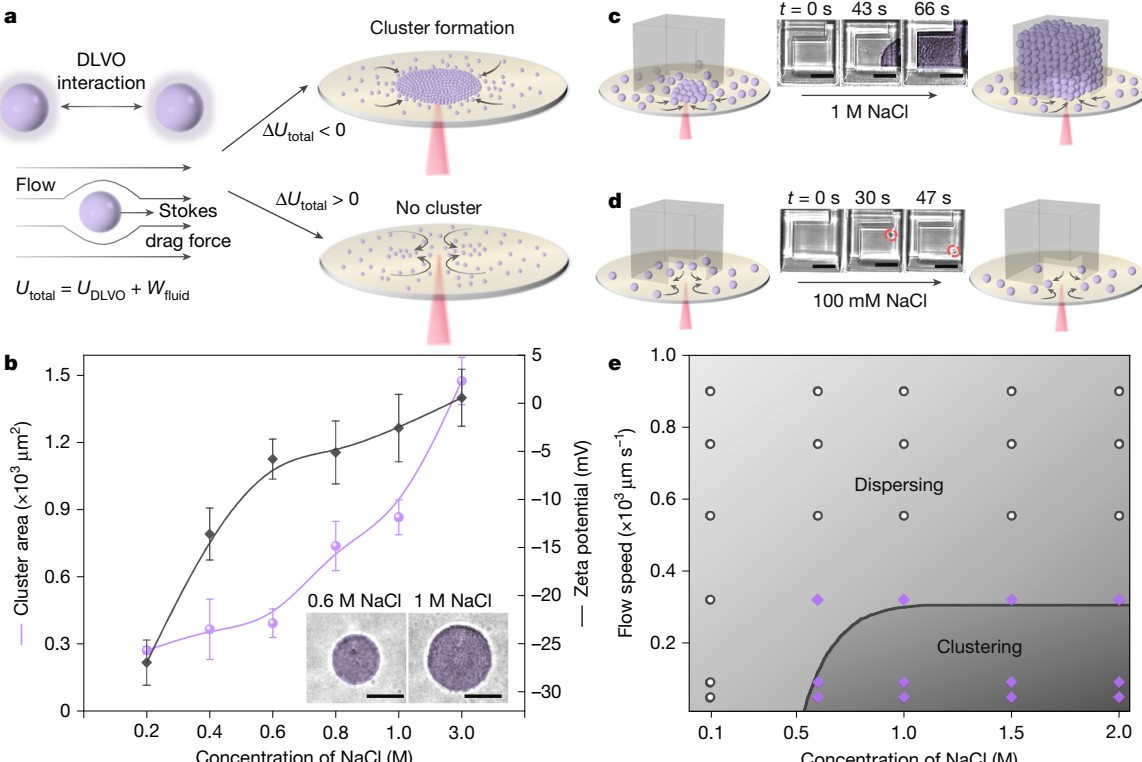

**Fig. 2 | Assembly mechanism. a**, Schematic illustration showing the competition between DLVO interactions and Stokes drag force and their influence on colloidal cluster formation following femtosecond laser heating. **b**, Cluster area (purple spheres) of $SiO_2$ particles dispersed in aqueous solutions with varying NaCl concentrations after 60 s of laser heating at a power of 50 mW and a scan speed of 5 μm s⁻¹, along with the corresponding zeta potential value (black diamonds) of these $SiO_2$ particles. The insets are optical images of $SiO_2$ clusters formed in 0.6 M and 1 M NaCl solutions. Data are presented as mean ± s.d. with at least four independent samples/measurements ($n ≥ 4$). **c,d**, Schematic illustrations and optical images depicting the assembly process of $SiO_2$ particles within a hollow microcube in 1 M (**c**) and 100 mM (**d**) NaCl solution. The microcube measures 32 μm in length and width and 10 μm in height, with an opening of 26 μm in length and 10 μm in height on the side. **e**, Theoretical phase diagram (grey background) and experimental results (open circles and solid diamonds) showing the influence of various concentrations of NaCl and flow speed on $SiO_2$ particle clustering. The light-grey region and open circles represent the domain in which $SiO_2$ particles remain dispersed and no cluster is formed, whereas the dark-grey region and solid diamonds represent the domain in which $SiO_2$ cluster formation occurs. Scale bars, 30 μm (**b**); 10 μm (**c,d**).

particle–particle attraction, leading to the clustering of the colloidal particles[45,46] (Fig. 3a). When hydrophilic $SiO_2$ particles are redispersed into hydrophobic solvents, such as immersion oil, mineral oil or oleic acid, the strong hydrophobic interaction facilitates continuous cluster growth (Fig. 3b), resulting in a faster growth of $SiO_2$ cluster under laser illumination (Fig. 3c). Consequently, $SiO_2$ particles can be efficiently assembled and fully filled within a hollow micro-template in these solvent systems (Supplementary Video 5). More importantly, the strong hydrophobic interaction enables efficient 3D assembly at a flow speed of several mm s⁻¹, even under substantial Stokes force, greatly improving assembly efficiency. Furthermore, the assembly speed can be modulated by adjusting the laser dosage (the laser power and scan speed; see Supplementary Fig. 7). Notably, $SiO_2$ particles dispersed in silicone oil exhibit poor clustering ability and fail to undergo 3D assembly within the hollow template (Supplementary Video 6). This might be because of weak attractive interaction among $SiO_2$ particles, as their strong affinity to silicone oil (attributing to their mutual Si–O bonds) prevents clustering.

Beyond solvent selection, surfactants can also enhance efficiency in 3D assembly in aqueous solutions. Laser illumination often induces bubble formation owing to high-temperature-driven evaporation, generating Marangoni flow caused by the surface tension gradients at the bubble–solution interface. Moderate flows promote $SiO_2$ particle clustering, whereas a strong flow inhibits cluster formation (Fig. 3d,e and Supplementary Video 7), as the bubble formation frequently results in intense fluid flow exceeding 4 mm s⁻¹. The addition of a surfactant,

such as hexadecyltrimethylammonium bromide (CTAB), effectively reduces the surface tension, limiting bubble growth and weakening the laser-induced Marangoni flow[39]. Moreover, the cationic ions (CTA⁺) dissociated from CTAB molecules absorb onto the surface of negatively charged $SiO_2$ particles, neutralizing their surface charge (Supplementary Fig. 8). This reduction in electrostatic repulsion enhances the DLVO attractions among the $SiO_2$ particles, thereby promoting their aggregation. As a result, the $SiO_2$ clustering is progressively enhanced by increasing CTAB concentration (Fig. 3e), enabling robust 3D assembly in 1 mM CTAB solution (Supplementary Video 8). Beyond CTAB (cationic surfactant), other surfactants, including anionic surfactant sodium dodecyl sulfate (SDS) and non-ionic surfactants such as polyethylene glycol (PEG) and Pluronic F-108 (PF108), exhibit similar effects on $SiO_2$ particle clustering (Fig. 3f). Notably, $SiO_2$ particles exhibit strong electrostatic repulsion in the solution of SDS (Supplementary Fig. 8), which may hinder the cluster formation. However, other inter-particle attractions, such as opto-thermophoretic force[23,34,47] and depletion force[48], may also arise in these surfactant systems, facilitating particle aggregation.

## Broad compatibility with versatile materials

Our strategy is superior at constructing complex 3D microstructures using a diverse range of micromaterials/nanomaterials, regardless of their shape, size and surface chemistry. To demonstrate the universality of this approach, we first assembled $SiO_2$ colloidal particles of

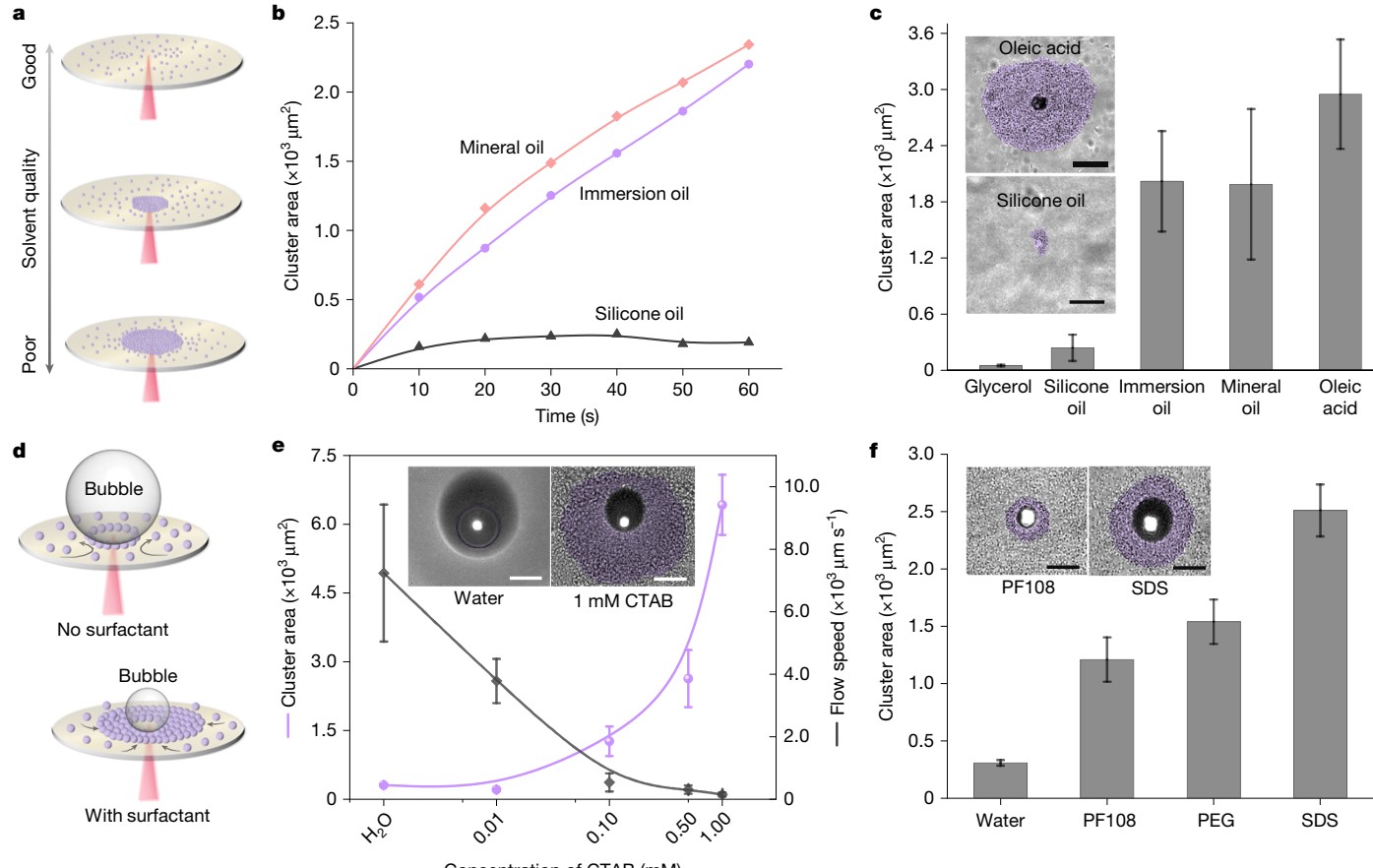

**Fig. 3 | Assembly optimization using various solvent systems. a**, Schematic illustration showing the clustering behaviours of colloidal particles in solutions with different particle–solvent affinities. **b**, Cluster area of SiO₂ particles over time in different solvents. Data points are extracted from Supplementary Video 16 at 10-s intervals. **c**, Cluster area of SiO₂ particles in various solvents after 60 s of laser heating at a power of 50 mW and a scan speed of 500 µm s⁻¹. Insets are optical images of SiO₂ clusters in oleic acid and silicone oil. **d**, Schematic illustration depicting the effect of surfactant on colloidal particle assembly in the aqueous solutions. The addition of surfactant reduces the surface tension gradient, attenuating the bubble growth and weakening Marangoni flow, thereby facilitating the particle clustering. **e**, Cluster area of SiO₂ particles (purple spheres) and flow speed (black diamonds) in aqueous solutions with different concentrations of CTAB. Insets are optical images of SiO₂ clusters in pure water and 1 mM CTAB solution. **f**, Cluster area of SiO₂ particles in solutions containing different surfactants (1 wt% PF108, 1 wt% PEG and 8 mM SDS). Insets are optical images of SiO₂ clusters in solutions of 1 wt% PF108 and 8 mM SDS. The laser power and scan speed for **e** and **f** are both 50 mW and 5 µm s⁻¹, respectively, with a duration of 10 s. Data points are shown as mean ± s.d. with at least three independent samples/measurements ($n \geq 3$). Scale bars, 30 µm (**c**,**e**,**f**).

varying sizes and surface chemistries into versatile 3D microstructures. These include a micro-gourd composed of 1-µm-diameter SiO₂ (Fig. 4a–c), a micro-hexagon made of 600-nm-diameter SiO₂ particles (Fig. 4d–f) and microcubes constructed from 1-µm-diameter green fluorescent SiO₂ particles (Supplementary Fig. 9) and 3-µm-diameter and 10-µm-diameter SiO₂ particles (Supplementary Video 9). Furthermore, particles of different sizes can be co-assembled to form heterogeneous 3D microarchitectures. For instance, a microsphere consisting of both 1-µm and 600-nm SiO₂ particles is successfully co-assembled (Fig. 4g–i), demonstrating the capability of integrating different components into 3D microstructures.

Furthermore, architectures at distinct locations can be site-selectively assembled with particles of different sizes or components by locally addressing the fs laser, with no cross-interference among these assembled architectures. For example, the alphabet letters 'P' and 'I' are sequentially assembled using 1-µm and 600-nm SiO₂ within a close mutual distance of about 10 µm on the substrate (Fig. 4j–l). Once assembled within a template, strong inter-particle interactions resist disturbances from the violent flow during sequential assembly process, preventing disassembly. This precise control enables the localized integration of several components for a targeted layout, paving the way for the fabrication of microdevices with spatially varying compositions and on-demand multifunctionalities (see details in Fig. 5). Beyond these micrometre-sized SiO₂ particles, our method is compatible with a broad range of nanomaterials, enabling the fabrication of diverse 3D structures with nanoscale features. As illustrated in Fig. 4m–r, a screw-like microstructure with helical threads of around 320 nm in width (Fig. 4m,n) and an alphabet letter 'E' with a height of around 855 nm (Fig. 4p–r) are created using TiO₂ nanoparticles with 90.2 ± 15.8 nm diameter and Fe₃O₄ nanoparticles with 16.7 ± 2.3 nm diameter, respectively. Notably, smaller nanoparticles with a uniform size distribution yield microstructures/nanostructures with smoother surfaces (Fig. 4q and Extended Data Fig. 1). Furthermore, as illustrated in Fig. 4s–z and Extended Data Fig. 2, we successfully assemble microcubes from various nanomaterials, including TiO₂ nanowires (NWs), diamond nanoparticles (Supplementary Fig. 10 and Supplementary Video 10), Fe₃O₄ nanoparticles, WO₃ NWs, Al₂O₃ NWs, Ag nanoparticles and CdTe quantum dots (Supplementary Video 11), respectively. Notably, although these architectures exhibit intrinsic mechanical robustness (as demonstrated by the dangling croissant-shaped superstructure in Fig. 1d,e) owing to strong intrinsic inter-particle interactions, post-treatments such as annealing can further promote intra-particle interfacial welding, substantially enhancing the mechanical properties of the 3D microarchitectures.

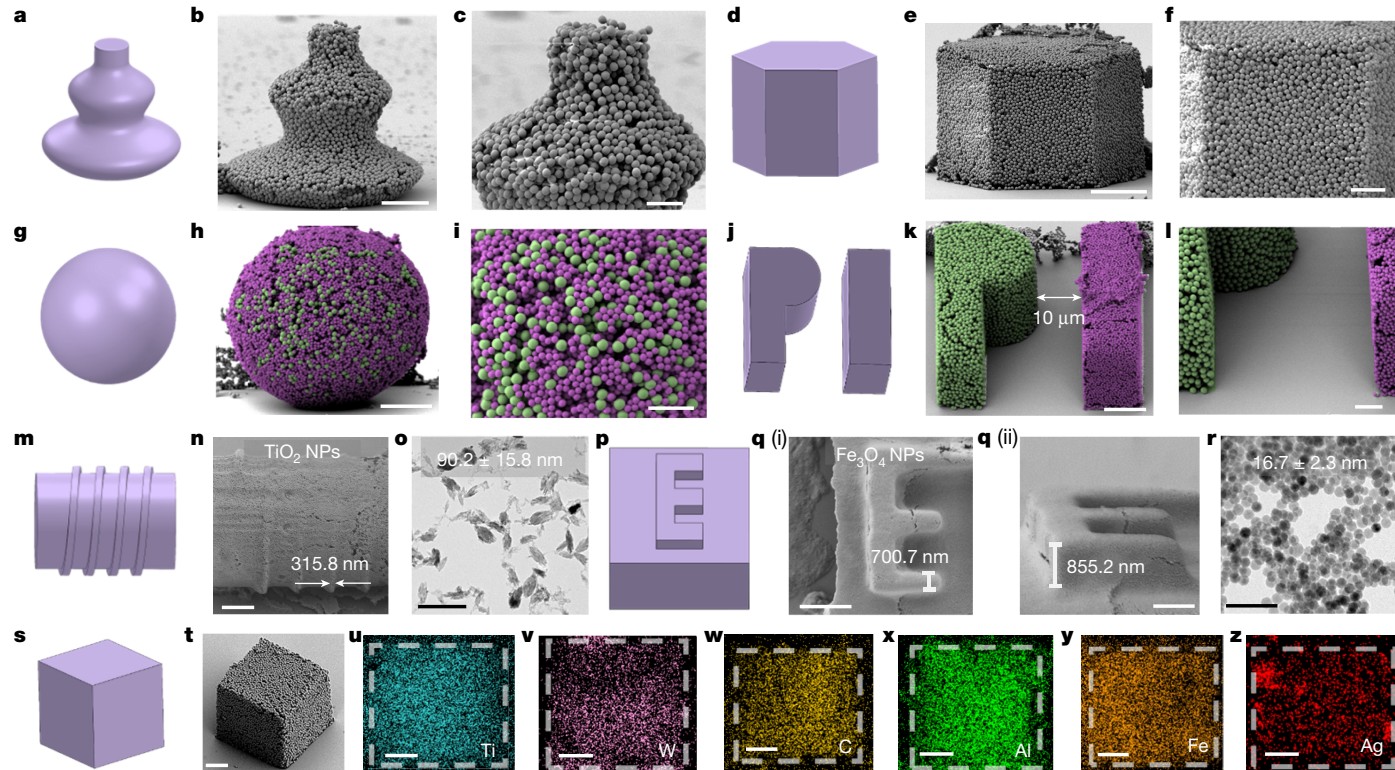

**Fig. 4 | Wide compatibility with versatile micromaterials/nanomaterials.** **a**–**c**, 3D model (**a**) and SEM images (**b**,**c**) of a micro-gourd superstructure assembled with 1-µm SiO₂ particles. **d**–**f**, 3D model (**d**) and SEM images (**e**,**f**) of a hexagon-shaped micro-superstructure assembled with 600-nm SiO₂ particles. **g**–**i**, 3D model (**g**) and SEM images (**h**,**i**) of a microsphere co-assembled with 1-µm and 600-nm SiO₂ particles. **j**–**l**, Model (**j**) and SEM images (**k**,**l**) of superstructures of the letters 'P' (assembled with 1-µm SiO₂ particles) and 'I' (assembled with 600-nm SiO₂ particles). **m**,**n**, 3D model (**m**) and SEM image (**n**) of a TiO₂ nanoparticle (NP)-assembled screw-like microstructure. **o**, TEM image of TiO₂ NPs. **p**,**q**, 3D model (**p**) and SEM images (**q** (i), (ii)) of the letter 'E' composed of Fe₃O₄ NPs. **r**, TEM image of Fe₃O₄ NPs. **s**–**z**, Model (**s**), SEM image (**t**) and EDS mapping (**u**–**z**) of a microcube assembled with various materials, including SiO₂ (**t**), TiO₂ NWs (**u**), WO₃ NWs (**v**), diamond NPs (**w**), Al₂O₃ NWs (**x**), Fe₃O₄ NPs (**y**) and Ag NPs (**z**). The high-resolution surface morphology of the microstructures assembled with different nanomaterials, along with the corresponding component nanomaterials, can be found in Extended Data Figs. 1 and 2. Scale bars, 10 µm (**b**,**e**,**h**,**k**,**t**); 4 µm (**c**,**f**,**i**,**l**); 5 µm (**u**–**z**); 2 µm (**n**,**q** (i)); 800 nm (**q** (ii)); 200 nm (**o**); 80 nm (**r**).

## On-demand creation of multifunctional microdevices

With the aid of precise spatial control and broad material applicability, microstructures spatially encoded with diverse functional materials can be created, highlighting the potential of our technique for developing microdevices with on-demand functionalities. As proof-of-concept demonstrations, microfluidic chips with tailored separation capabilities for tiny objects are first demonstrated (Fig. 5a–e). This microfluidic chip consists of a polymeric microchannel printed by the 2PP process with a colloidal-particle-assembled microvalve embedded inside (Fig. 5b,c). The microvalve, measuring 40 µm in length and 20 µm in width, is entirely assembled from 1-µm SiO₂ particles, forming complex 3D porous channels (Fig. 5c). When a droplet of nanoparticle suspension is introduced at one end of the microfluidic chip, capillary forces instantaneously drive the solution into the microchannel (Fig. 5a). The colloidal microvalve, with feature porosity of several hundred nanometres, permits the rapid permeation of solvent flow in seconds (Fig. 5b), while efficiently rejecting and retaining nanoparticles (Fig. 5d). As solvent evaporation continues at the opposite end of the chip, nanoparticles progressively accumulate at the inlet side of the microvalves, forming an enrichment zone (Fig. 5a and Supplementary Fig. 11), which holds potential for trace substance detection. By modulating the size of the microvalves, we can selectively separate particles of different sizes. For instance, 100-nm poly(lactic-*co*-glycolic) acid (PLGA) nanoparticles are successfully sorted out using a SiO₂ valve of 40 µm in both width and length (Fig. 5e). Furthermore, we also construct a microfluidic chip

embedded with several concatenated microvalves composed of different materials, each featuring distinct porosities and cut-off capabilities, achieving the size-selective sorting of a mixture of different particles (Supplementary Fig. 12).

Subsequently, we demonstrate the application of our technique in fabricating multifield-driven microrobots with multimodal locomotion (Fig. 5f–s). As illustrated in Fig. 5f,g, a Fe₃O₄ nanoparticle-assembled cylinder microrobot exhibits magnetically controlled tumbling on the substrate (Supplementary Video 12). The magnetic response can be flexibly tuned by regulating the Fe₃O₄ nanoparticle content (Supplementary Fig. 13). Beyond magnetic actuation, Fig. 5h–o illustrates light-driven microrobots with controlled motion enabled by tailoring both the shape and spatial distribution of functional materials. For example, a cylinder heterojunction TiO₂–Au microrobot demonstrates linear propulsion through self-electrophoresis[49] under ultraviolet (UV) illumination in water (Fig. 5h,i, Supplementary Fig. 14 and Supplementary Video 13, part 1). Shape asymmetry further enables rotational motion: a T-shaped TiO₂–Au microrobot rotates in tight circles (Fig. 5j,k, Supplementary Video 13, part 2), whereas a V-shaped design produces larger circular trajectories (Fig. 5l,m and Supplementary Video 13, part 3). Moreover, alternative spatial encoding of functional materials allows the same V-shaped microrobot to switch from rotation to linear motion (Fig. 5n,o and Supplementary Video 13, part 4). Notably, our technique also enables the integration of several functional materials into a single microrobot, achieving multistimulus responsiveness. For instance, we fabricate an

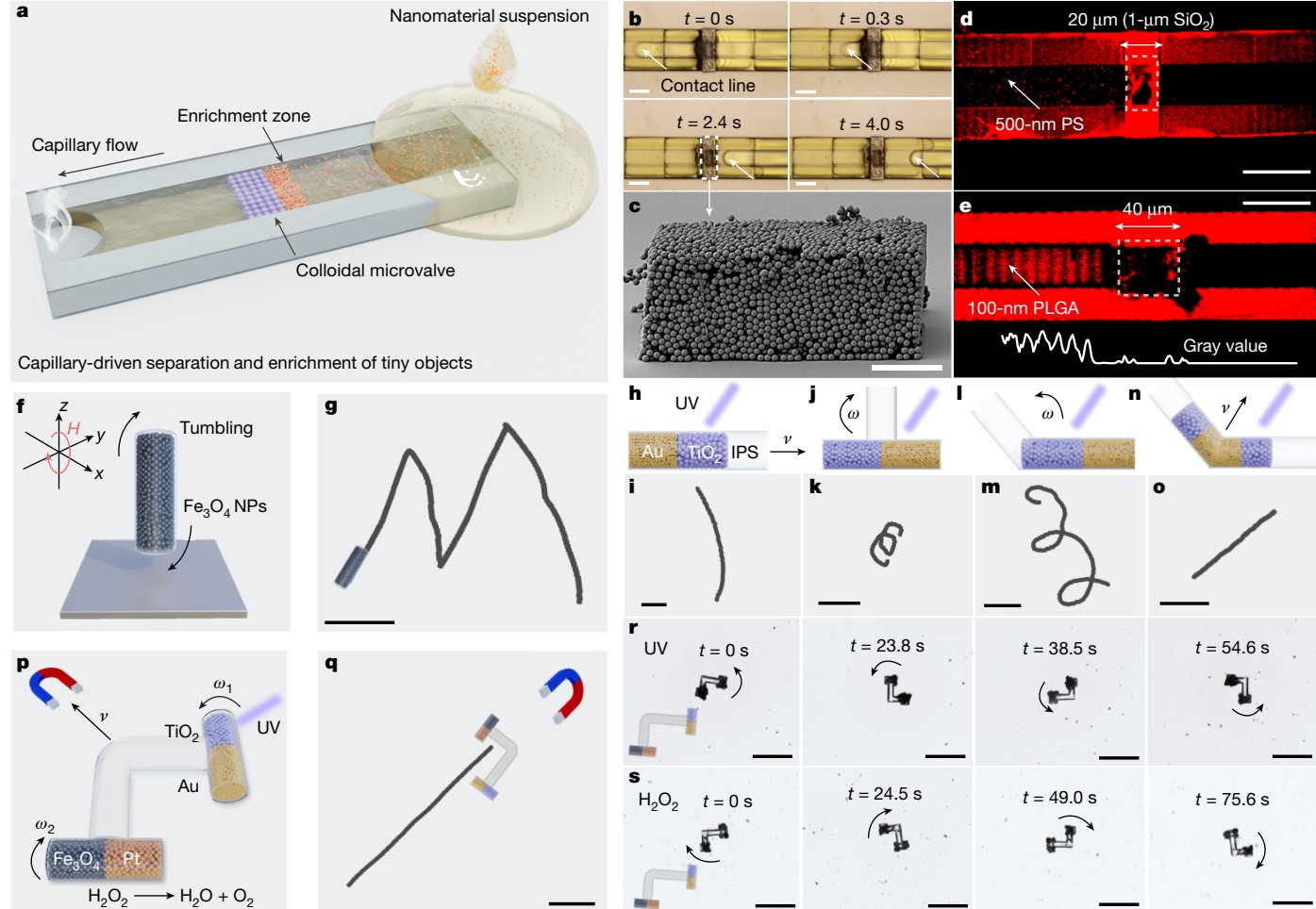

**Fig. 5 | On-demand construction of multifunctional microdevices.**
**a–e**, Microfluidic sieving devices. **a**, Schematic illustration showing the
capillary-driven separation process of a microfluidic sieving device. **b**, Time-
lapse optical images showing the capillary-driven fluid flow through a
microfluidic chip embedded with a colloidal microvalve (20 μm width,
40 μm length) composed of 1-μm $SiO_2$ particles. Images are extracted from
Supplementary Video 17. **c**, SEM image of the $SiO_2$-assembled microvalve.
**d**, Fluorescent image of 500-nm polystyrene (PS) nanospheres rejected by
the microvalve. **e**, Fluorescent image showing 100-nm PLGA nanoparticles
rejected by the microvalve. Inset is the variation of gray value along the
microchannel. **f–s**, Multifield-driven microrobots. **f,g**, Schematic illustration
(**f**) and trajectory (**g**) of the magnetic tumbling of a $Fe_3O_4$ nanoparticle (NP)-
assembled cylinder microrobot. **h–o**, Schematic illustrations (**h,j,l,n**) and
motion trajectories (**i,k,m,o**) of $TiO_2$–Au microrobots with different shapes
and material distributions. **p–s**, Schematic illustration (**p**) of an L-shaped
microrobot integrated with four different materials ($TiO_2$, Au, Pt and $Fe_3O_4$)
and its three motion modes: magnetic pulling (**q**), light-driven anticlockwise
rotation (**r**) and clockwise rotation in about 5 wt% $H_2O_2$ (**s**). Scale bars, 50 μm
(**b,d,e**); 60 μm (**g,i,k,m,o,r,s**); 30 μm (**q**); 10 μm (**c**).

L-shaped microrobot incorporating Au, $TiO_2$, Pt and $Fe_3O_4$ (Fig. 5p–s),
which exhibits three distinct motion modes: magnetic pulling, anti-
clockwise rotation under UV light and clockwise rotation in $H_2O_2$
(Supplementary Video 14).

In conclusion, our technique overcomes the material limitation in
conventional 3D microfabrication/nanofabrication, enabling the crea-
tion of complex, volumetric 3D microstructures/nanostructures from
a diverse range of materials. We believe that this step change in fab-
rication capability—bridging optical physics, colloidal science, fluid
mechanics and device engineering—not only provides new insights
into fundamental colloidal assembly but also opens new frontiers
in various fields, such as reconfigurable photonics, multifunctional
microdevices/microrobots and biologically integrated systems.

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

## Methods

### Materials

$SiO_2$ colloidal particles with various sizes (140 nm to 5 µm, 5 wt%), green-fluorescent $SiO_2$ (1 µm, excitation/emission: 497/530 nm, 2.5 wt%) and red-fluorescent polystyrene nanospheres (500 nm and 2.5 µm, excitation/emission: 530/607 nm, 2.5 wt%) were purchased from microParticles GmbH. Mineral Oil Rotational Viscosity Standard (494.0 mPa s), oleic acid (technical grade, 90%), silicone oil AR20 (viscosity: approximately 20 mPa s), CTAB (98%), SDS (98.5%), PEG (Mn: 400), PF108 (Mn: 14,600), $TiO_2$ NWs (powder, 100 nm × 10 µm), $WO_3$ NWs (powder, 50 nm × 10 µm), $Al_2O_3$ NWs (powder, 2–6 nm × 200–400 nm), CdTe quantum dots (powder, COOH functionalized, fluorescent emission: 710 nm), iron oxide ($Fe_3O_4$) powder (50–100 nm, 97%), $Fe_3O_4$ (20 nm, 5 mg ml$^{-1}$, dispersed in $H_2O$), $TiO_2$ nanoparticles (150 nm, 900 nm, 5 wt%, dispersed in $H_2O$), Au urchin nanoparticles (50 nm, in 0.1 mM PBS), Pt nanoparticles (powder, 50 nm, 99.9%), silver (Ag) powder (<100 nm, PVP as dispersant, 99.5%), orange fluorescent PLGA nanospheres (excitation/emission: 530/582 nm, 100 nm), diamond nanopowder (<10 nm, 97%), trichloro(1H,1H,2H,2H-perfluorooctyl) silane (97%), sodium chloride (NaCl, 99%), $H_2O_2$ (30% w/w in $H_2O$) and propylene glycol monomethyl ether acetate (PGMEA, 99.5%) were purchased from Sigma-Aldrich. Isopropyl alcohol (IPA, 99.9%) was purchased from Carl Roth GmbH. IPS photoresist was purchased from Nanoscribe GmbH. Immersion oil 518 F (viscosity of about 500 mPa s) was purchased from Fisher Scientific.

### Fabrication of 3D hollow microtemplates

The 3D hollow polymeric templates were designed using SOLIDWORKS 2021 and fabricated by direct laser writing using a commercial Photonic Professional GT system (Nanoscribe GmbH). The templates were printed with commercial IPS photoresist in oil-immersion mode with a 63× objective and printed on a fused glass substrate that had been treated overnight with trichloro(1H,1H,2H,2H-perfluorooctyl) silane vapour. The laser power and printing speed were 50 mW and 50,000 µm s$^{-1}$, respectively. After printing, the structures were developed in IPA solution for 5 min to remove non-polymerized photoresist. The templates were then directly used for the particle assembly in particle-laden dispersions. For aqueous dispersion, the high thermal conductivity of water can limit photothermal efficiency. To enhance this, a 5-nm Cr layer followed by a 5-nm Au layer was sputtered onto the substrate, improving photothermal performance.

### Process of the optofluidic 3D microfabrication/nanofabrication

$SiO_2$ particles of different diameters (150 nm and 1 µm) were used in the assembly experiments. To redisperse $SiO_2$ particles into various solvent systems, 100 µl of as-received aqueous $SiO_2$ solution was first centrifuged at 8,000 rpm for 3 min. The supernatant was discarded and the collected $SiO_2$ precipitate was dried by heating at 60 °C for 2 h. The dried $SiO_2$ powder was then redispersed in 1 ml of various solvents (for example, immersion oil, oleic acid) and sonicated for 1 h to ensure thorough dispersion. These hydrophilic $SiO_2$ particles can remain dispersed in various oils in the experimental timescale owing to the high viscosity of the medium, which helps prevent clogging the template openings by large agglomerations during assembly (Supplementary Fig. 15). The resulting $SiO_2$ suspension was used for the assembly experiments. It should be noted that other colloidal materials were also redispersed in appropriate solvents using a similar procedure to perform the assembly experiments. Furthermore, co-assembly of different materials can be obtained by preparing a mixed suspension through physical blending of various particles, as demonstrated in Fig. 4i, in which $SiO_2$ microspheres with 600 nm and 1 µm diameter were co-assembled into a single structure. The detailed parameters of colloidal materials used in this study are summarized in Supplementary Table 3.

For experiments conducted in aqueous solutions with varying NaCl concentrations, 150-nm $SiO_2$ particles were selected because of their reduced gravitational settling, ensuring system stability before laser-induced assembly. $SiO_2$ dispersions were prepared with different NaCl concentrations at a particle concentration of 1 wt%. To study assembly dynamics and ensure clear observation, large $SiO_2$ particles (1 µm) were used in various solvent systems, including aqueous solutions containing surfactants and oil-based solvents. A 100-µl aliquot of each prepared solution was placed inside a square PDMS spacer (1 cm side length, about 200 µm height). The spacer was sealed with a cover glass to prevent evaporation. A fs laser beam (780 nm, 80 MHz) with a scanning area of 2 × 2 µm$^2$ was continuously applied at the corner of the printed hollow template to guide particle assembly. After the assembly process, the substrate was sequentially washed with IPA and deionized water, followed by mild sonication to remove any residual colloidal particles surrounding the microstructure.

To create 3D structures or devices made from different particles, we used a sequential, multistep assembly procedure. Suspensions of different particles were introduced successively, with a washing step between each stage to remove excess particles and prevent cross-interference. For instance, when creating alphabet letters 'P' and 'I', a suspension of 600-nm $SiO_2$ was first used to assemble 'I'. The excess suspension was then rinsed away with IPA and water to ensure clean conditions for the next step, after which a suspension of 1-µm $SiO_2$ particles was introduced to assemble 'P'. By repeating these processes, several particle systems can be used to construct distinct structures at predefined locations on a single substrate or to achieve seamless integration within a single microstructure, thereby enabling the creation of multifunctional microdevices.

Finally, it should be noted that excessively strong inter-particle attraction can lead to the formation of large clusters that clog the openings, preventing complete filling of the 3D structures (Supplementary Video 15). This issue can be mitigated by optimizing the size and distribution of template openings, as shown in Supplementary Discussion 1 and Supplementary Figs. 16–18.

### Template removal

To completely remove the outer-layer polymer template and maintain the structural integrity, the colloidal-assembled microstructures were subjected to mild $O_2$ plasma treatment for 6 h with a plasma cleaner machine (Tergeo-EM, PIE Scientific) at 75 W power and an $O_2$ flow rate of 20 sccm. To further enhance interfacial bonding, the microstructures were annealed in air at 600 °C for 2 h with a heating rate of 2 °C min$^{-1}$. For faster template removal, a high-power Plasma Asher (MPR-6) can also be used.

### Fabrication of microfluidic chips for particle separation

The microfluidic channel was fabricated using the Nanoscribe system with IPG-780 photoresist. To enhance fabrication efficiency, template models with an asymmetric hole configuration were designed, similar to the one shown in Supplementary Fig. 10. This design enabled direct printing of the microchannel without requiring polymer template removal after particle assembly. Specifically, 100 µl of IPG-780 was dispensed onto the substrate and pre-baked at 100 °C for 1 h. A microchannel (2 mm length, 40 µm width, 15 µm height) was then printed using the Nanoscribe system with a 63× objective. The laser power and printing speed were set to 30 mW and 80,000 µm s$^{-1}$, respectively. After printing, the substrate was developed in PGMEA for 20 min, followed by IPA for 1 min to remove unpolymerized photoresist. For the microfluidic separation experiments, colloidal particle solutions containing 30 v/v% IPA were prepared to improve wettability within the microchannel.

### Microrobot motion experiments

The fabricated microrobots were first treated with $O_2$ plasma for 5 min to change their wettability before use. Afterwards, these microrobots

were scratched off the glass and dispersed in deionized water. The microrobot suspension was then added to a piece of glass slide and observed under an optical microscope (ZEISS Axio Imager 2). The magnetically actuated tumbling was conducted under a uniform rotating field with an amplitude of 10 mT and frequency of 1–120 Hz. The light-driven motion is powered by a UV light-emitting diode with a wavelength of 365 nm (Thorlabs).

## Microscopy characterization

The real-time particle assembly process was recorded using a built-in AxioVision Live-View camera in the Nanoscribe system. The recorded videos were analysed using the Manual Tracking plugin in Fiji software to determine the flow speed. The microfluidic separation experiments were recorded using an optical microscope (ZEISS Axio Imager 2). The SEM and energy-dispersive X-ray spectroscopy (EDS) images were obtained using a Gemini field-emission SEM 500 at an acceleration voltage of 5 or 20 kV. The transmission electron microscopy (TEM) images were acquired using a Philips CM200 TEM with an accelerating voltage of 200 kV. The fluorescent images were captured by a SP8 Leica confocal microscope and the raw data were processed and colourized using the built-in software LAX. The excitation wavelength was set to 552 nm with 10% intensity and the emission wavelength was collected at 630–700 nm.

## Zeta potential measurement

The zeta potential of $SiO_2$ (140 nm and 1 μm) colloidal particles dispersed in various solvents was measured in a Zetasizer Ultra system (Malvern). The viscosity and relative permittivity of solutions were set at $8.9 \times 10^{-4}$ Pa s and 78.5, respectively. The final zeta potential of $SiO_2$ particles was obtained by averaging the results of five independent measurements.

## Finite-element simulations

A 3D model was built with the COMSOL Multiphysics package (version 6.0) to simulate the fluid flow around a 2PP-printed hollow microcube. The size of the hollow cube was $40 \times 40 \times 40$ μm³ with a hole of $10 \times 10 \times 10$ μm³ at one of its corners (Supplementary Fig. 19). The computational domain was a cubic box of $400 \times 400 \times 400$ μm³, with the upper surface set as a free boundary at a temperature of $T_0 = 293.5$ K. The four side boundaries were modelled with symmetric boundary conditions, whereas the remaining boundaries were set as no-slip conditions. A laser with a scanning area of $2 \times 2$ μm² was applied to reach $T_1 = 2{,}293$ K. The box was filled with water and the density variation induced by heat transfer drove the flow through buoyancy. The temperature distribution and fluid flow field surrounding the cube were computed using the laminar flow and conjugate heat transfer modules within the box.

## Theoretical analysis

To explain the competition between inter-particle interactions and particle–fluid interactions during the assembly process, a model system in which $SiO_2$ particles assemble in aqueous solutions with various NaCl concentrations was used. On the basis of our experimental observations (Fig. 2b), the particle suspension initially resided in a metastable dispersion state, in which the attractive van der Waals interaction outweighed the electrostatic repulsion. In this system, the work done by the particle–fluid interaction during an infinitesimal displacement $ds$ is represented by

$$\Delta W_{fluid} = 6\pi \mu R v \times ds \tag{1}$$

in which $\mu$ is the dynamic viscosity of the fluid, $R$ is the particle radius and $v$ is the flow velocity.

The difference in interaction potential between particles is represented by

$$\Delta U_{DLVO} = F_{DLVO} \times ds. \tag{2}$$

DLVO force ($F_{DLVO}$) consists of two parts: van der Waals force (attractive force for the formation of colloidal clusters) and EDL repulsion force among particles.

The van der Waals force between particle and particle ($F_{vdW-pp}$) can be described by

$$F_{vdW-pp}(h) = \frac{A}{6}\left(-\frac{4R^2(2R+h)}{[(4R+h)h]^2} - \frac{4R^2}{(h+2R)^2} + \frac{8R^2}{h(4R+h)(h+2R)}\right) \tag{3}$$

in which $A$ is the Hamaker constant and $h$ is the closest distance to the two-particle surface.

The EDL energy[50] between particle and particle can be described by

$$W_{EDL-pp}(h) = \pi\varepsilon_0\varepsilon_l R\zeta^2\left[\ln(1 - e^{-2\kappa h}) + \ln\left(\frac{1 + e^{-\kappa h}}{1 - e^{-\kappa h}}\right)\right] \tag{4}$$

in which $\zeta$ is the zeta potential on the solid–fluid interface, $\varepsilon_0$ and $\varepsilon_l$ are the dielectric permittivities of vacuum and water, respectively, $\kappa^{-1} = \sqrt{\frac{\varepsilon_0\varepsilon_l K_B T}{2N_A e^2 I}}$ is the Debye length, $I = 0.5 \sum z_i^2 \rho_i$, $\rho_i$ is the ion concentration and $z_i$ is the number of charges carried by the ion.

The EDL repulsion force ($F_{EDL-pp}$) is then described by

$$F_{EDL-pp}(h) = -\frac{\partial W_{EDL-pp}(h)}{\partial h} \tag{5}$$

$$F_{EDL-pp}(h) = -2\pi\varepsilon_0\varepsilon_l R\kappa\zeta^2\left[\frac{e^{-2\kappa h}}{1 - e^{-2\kappa h}} - \frac{e^{-\kappa h}}{1 - e^{-2\kappa h}}\right] \tag{6}$$

in which $h \to 0$

$$\lim_{h \to 0} F_{EDL-pp}(h) = \pi\varepsilon_0\varepsilon_l R\kappa 2\zeta^2\frac{3\kappa h}{2\kappa h} = 3\pi\varepsilon_0\varepsilon_l R\kappa\zeta^2 \tag{7}$$

Then the $\Delta U_{total}$ can be described by

$$\Delta U_{total} = (F_{EDL-pp} + F_{vdW-pp} + 6\pi\mu R v) \times ds \tag{8}$$

When increasing the ionic strength of the system, the $F_{EDL-pp}$ is gradually screened. Therefore, colloidal particle clustering becomes more favourable.

The tendency of the system to form clusters can then be evaluated by examining changes in $U_{total}$. When $\Delta U_{total} < 0$, the attractive interaction dominates in the fluid flow, leading to the cluster formation of colloidal particles; otherwise, the repulsive interaction dominates and particles would remain dispersed. The detailed information of all physical parameters used in this study, including their corresponding symbols and values, are summarized in Supplementary Table 4. The phase diagram in Fig. 2e showing the dispersing and clustering domains of colloidal particles can be obtained with equation (8) through MATLAB.

## Data availability

All data are available in the main text or the Supplementary Information. Source data are provided with this paper.

50. Hogg, R., Healy, T. W. & Fuerstenau, D. W. Mutual coagulation of colloidal dispersions. *Trans. Faraday Soc.* **62**, 1638–1651 (1966).

**Acknowledgements** We acknowledge support from the Max Planck Society, Chinese Scholarship Council (X.L.), Alexander von Humboldt Foundation (M.Z.), European Research Council (ERC) Advanced Grant SoMMoR project with grant no. 834531 (M.S.), the Knut and Alice Wallenberg Foundation for the Wallenberg Initiative Materials Science for Sustainability (WISE), ERC under LUBFLOW (ERC-CoG-101088639) grant (W.L. and S.B.) and National

University of Singapore Presidential Young Professorship Start Up (A-0010080-00-00) Grant (M.Z.). We thank Y. Yu, Y. Hou, M. Kelsch and A. Shiva for their technical assistance.

**Author contributions** M.Z. conceived the idea, proposed the research and conducted the initial experiments to validate the concept. X.L., M.Z. and M.S. designed and planned the research. X.L. performed the experiments, with technical assistance from W.L., G.G., M.T.A.K. and M.Z. W.L. and S.B. carried out the theoretical analysis and simulations. X.L. analysed the data. M.Z. and M.S. supervised the research. X.L. and M.Z. wrote the original draft, with edits from all authors.

**Funding** Open access funding provided by Max Planck Society.

**Competing interests** The authors declare no competing interests.

**Additional information**
**Correspondence and requests for materials** should be addressed to Mingchao Zhang or Metin Sitti.

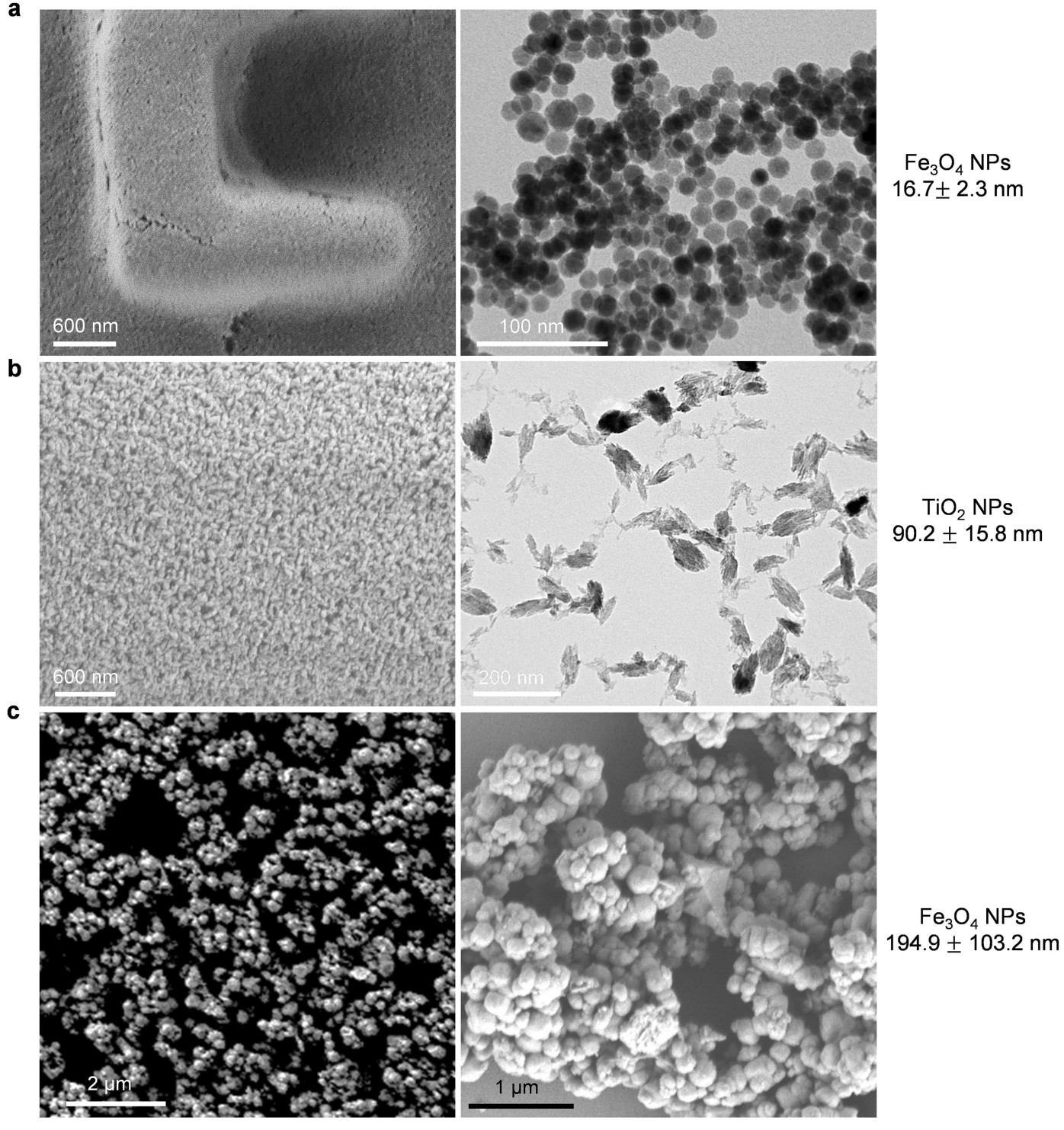

Fe$_3$O$_4$ NPs
16.7 ± 2.3 nm

TiO$_2$ NPs
90.2 ± 15.8 nm

Fe$_3$O$_4$ NPs
194.9 ± 103.2 nm

**Extended Data Fig. 1 | Surface morphology of microstructures assembled from TiO$_2$ nanoparticles and Fe$_3$O$_4$ nanoparticles with different size distribution. a**, SEM image (left) of the zoomed-in surface of the alphabet letter 'E' assembled from Fe$_3$O$_4$ nanoparticles (NPs) with a uniform size distribution (16.7 ± 2.3 nm) and TEM image (right) of the corresponding NPs. **b**, SEM image (left) of the zoomed-in surface of a cylinder assembled from TiO$_2$ NPs with a size distribution of 90.2 ± 15.8 nm and TEM image (right) of the corresponding NPs. **c**, SEM image (left) of the zoomed-in surface of a microcube assembled from Fe$_3$O$_4$ NPs with a broad size distribution (194.9 ± 103.2 nm) and SEM image (right) of the corresponding NPs.

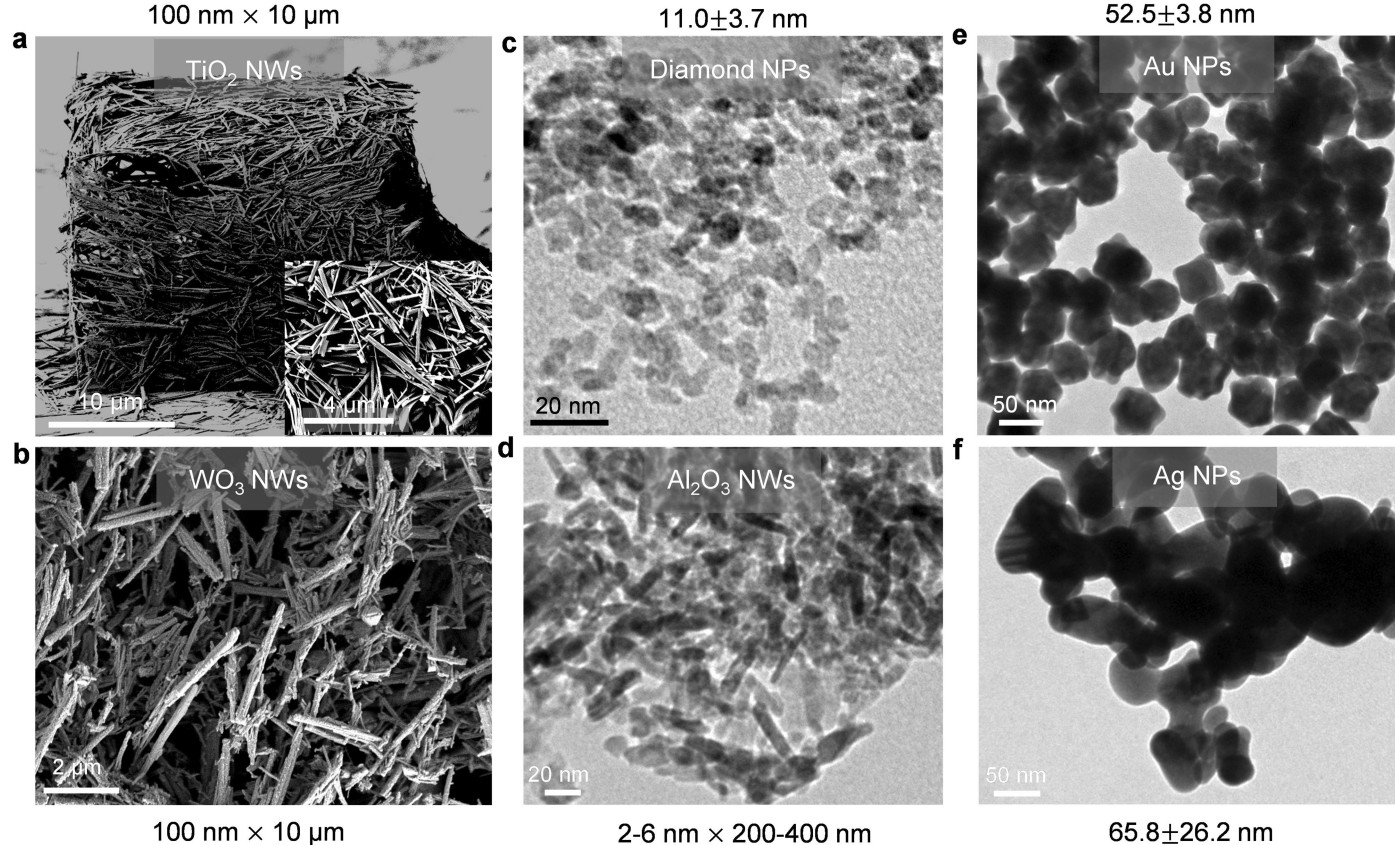

**Extended Data Fig. 2 | Morphology of various nanomaterials used in this work. a**, SEM image of a microcube assembled with $TiO_2$ NWs. Inset is the SEM image of $TiO_2$ NWs. **b**, SEM image of $WO_3$ NWs. **c**–**f**, TEM images of diamond nanoparticles (NPs) (**c**), $Al_2O_3$ NWs (**d**), Au NPs (**e**) and Ag NPs (**f**).