## [Peer Review file · Nature]

Optofluidic three-dimensional micro- and nanofabrication

Corresponding Author: Professor Metin Sitti

Version 0:

Reviewer comments:

Referee #1

(Remarks to the Author)

The manuscript presents results, analysis and demonstration of an “optofluidic” microfabrication method. Particles of various types are manipulated by using laser light and assemble inside templated shapes or within microfluidic devices. The work is of high scientific quality, and the manuscript includes excellent analysis of some of the key principles involved in the process based on colloidal stability theory. The results are very well illustrated, and the corresponding movies are impressive. Without doubt, this is a high-quality professional paper. However, it has some drawbacks when considered for a publication in a very high impact journal:

(1) The optofluidic method presented in the paper appears to be a modification of the optical thermophoresis patterning techniques that have attracted large attention recently and have been subjects of significant research, including in more mainstream publications. For an example of recent perspective see: <https://pubs.acs.org/doi/10.1021/acs.langmuir.5c01023>. The current paper extends the physics and application area of this family of methods, but does not present in my view principally new advances that could not have been foreseen based on others' recent progress.

(2) The introduction of the manuscript presents the technique as an alternative to the common 2PP techniques. However, the implementation of the method to make the 3D structures presented here relies on first using 2PP for making the templates where the particles would be driven and assembled optofluidically.

(3) The paper claims that the optofluidic technique can be applied to biological systems. However, the use of laser heating, even in scanning mode, is likely to lead to damage in delicate biomaterials and living systems. One element that would have provided useful information is the evaluation of the thermal impact and possible damage in biomolecular structures.

(4) The method is touted as an advancement in “nanofabrication”, yet it has been applied to particles that are closer to the microscale than the nanoscale. Some of the results, as the microsieving in Fig. 5 and the materials scale shown in Figure 5g apply to the ability of any porous microstructure to capture objects in its matrix. Fig. 5g seems to imply that the method uses new principles of manipulating the objects shown, which is misleading.

In summary, this is a paper that is rich in detail, demonstration of microfabricated structures and investigation and analysis of the basic colloidal stability phenomena involved in the optofluidic assembly process. The capabilities of the technique are demonstrated with numerous common types of particles and some of the structures and dynamics are quite impressive visually. However, the underlying optothermal manipulation method has already been studied elsewhere and has inherent limitations. Thus, while this paper would make an excellent contribution to an above-average impact journal, I am not very impressed by its level of novelty and importance as compared to the extremely high expectations for publication in Nature.

Referee #2

(Remarks to the Author)

Creating free-form objects at smallest dimensions remains challenging. Despite intense developments in additive manufacturing techniques, the state-of-the-art processes (2-photon polymerization) are limited by a selected choice of

(mostly polymeric) resins. Creating similar structures from functional materials may unlock a wide range of application in miniaturized devices, from chromatography and detection to nanophotonics and plasmonics. In their manuscript, Lyu et al. pick up this challenge and develop a bottom-up manufacturing process that delivers such functional, freeform 3D nanostructures from colloidal building blocks. The process uses polymeric 3D printed objects as templates and delivers particles within this cavity by optofluidic flow fields created by a femtosecond laser. They elucidate process conditions that trigger nucleation and cluster growth within these templates, until eventually, the object is completely filled and can be released by decomposing the template.

The process described in this well-written and appealing manuscript is innovative, thoroughly characterized and the authors carefully elucidate the underlying physics. The process pushes the limits of nanofabrication and provides a new paradigm for the design of tailored functional, hybrid, and nanoporous materials at the smallest scale with a wide range of applications. In my view, the manuscript therefore appeals to a broadest community of readers at the intersection of science, technology, and manufacturing and is thus of interest for Nature. The following points may serve to further improve on the description and understanding of the manuscript:

1) Mechanism

- Why is the templating form needed at all? It seems that for fundamental studies of the clustering threshold (e.g. Figure 2), the authors did not use any template at all and the process simply operates by triggering clustering via overcoming the energy balance towards aggregation. I assume that the template helps keep a defined form (instead of a freeform “writing” process with less defined edges and symmetry), but this could be more clearly discussed in the manuscript.
- When particles aggregate via vdW forces upon “delivery” via the optofluidic flow by overcoming electrostatic repulsion, why do they not directly clog the opening into the template? How can it be ensured that the entire hollow template fills up with particles efficiently (which seems to be the case in the objects the authors printed)?
- In the (co)assembly of binary mixtures, the authors demonstrate that they can either merge both populations, or create pure structures from one and the other particle system, respectively. What determines which of the two cases occurs? I assume that different particle systems exhibit different tendencies to cluster (i.e. have a different clustering threshold). Is that correct and if so, there should be criteria that one object needs to be printed first (i.e. of the easier-to-cluster population). A little more detail here would be useful.
- Why do the particles seemingly assemble in a well-ordered form (or is that impression of the images, e.g. Fig 1 misleading), if they are produced by a diffusion-limited aggregation process? In my view, to create ordered structures, the particles need to be able to probe the energy landscape at least to a certain extent (i.e. need to have sufficient time to assemble). Maybe the authors could comment on this aspect a bit.

2) Boundary conditions:

- It would add to the manuscript if the authors discuss some boundary conditions of the process. These could include: i) the critical size of the hole (maybe with respect to the entire template, or the particle size used to assemble the freeform); ii) the sizes of the particles that can be used (can they be too small or too big, and if so, why); iii) the laser energy (can the bubble from overheating be too large?); and iv) the obtainable size of the structures (will crack formation occur upon a critical size)? This would be useful for a reader to fairly assess how to apply the process to his/her needs and requirements.

3) Details:

- The description of the role of surfactants seems a bit oversimplified. On page 6, the authors mention that “surfactants introduce [...] enhanced DLVO force”, further promoting colloidal aggregation”. The behavior very much depends on the nature of the surfactant. In their case, a negatively-charged silica particle will indeed be destabilized by the addition of CTAB, but not because of it being a surfactant per se, but because of the positive charge. SDS for example would likely promote stability and thus hinder the process.
- On P7 and in Figure 4, the authors demonstrate the use of a range of particle systems. In some cases, the order and homogeneity of the print seems to be markedly worse (e.g. Fe₃O₄ or Ag; Figure 4q,w). Why is that? For the case of Ag nanoparticles, are localized surface plasmon resonances excited with the laser light NPs and does the resulting plasmonic heating interfere with the process (or can it be used to make the process more efficient)?
- How are hydrophilic SiO₂ particles dispersed in oil? In my experience, such systems will not be colloidally stable, but remain agglomerated. Did the authors characterize the size-distribution?
- In the experimental section, the authors mention that they apply oxygen plasma for 6h. Is that really needed and would the subsequent annealing step not suffice to consolidate the structures?

Referee #3

(Remarks to the Author)

In this paper the authors present a 3D Optofluidic “Nanofabrication” technique and demonstrate the assembly of different shapes, an assembled filtration system, and perform an analysis on the mechanism. The technique relies on creating a “template” of sorts using two photon polymerization and then filling the cavity with beads using optically driven thermal convective flow. As noted, a variety of shapes are created and a fairly substantive analysis is performed to understand some of the basic physics behind the process.

Ultimately though I think the contribution here falls below the standard of what I would expect for a publication in Nature – primarily on the basis of novelty. A few reasons

- 1) Optically driven thermal flows and particle concentration has been demonstrated many times in the past and similar, perhaps more extensive, analysis has been performed.

- 2) Ultimately this is just using that technique to fill up a cavity that is created by 2 photon polymerization. Given that this is required (or a similar technique) to create truly interesting shapes – its unclear to me why you couldn't just create the shapes using 2PP to begin with. Arguments in the paper are not convincing.
- 3) The demonstrated application of assembly of particle filters within a microfluidic chip has been demonstrated previously and the application itself is not very interesting. They discuss filtration of viruses and much smaller particles but only show a very simple demonstration.
- 4) Finally this is a very generous use of the term nanofabrication. While there may be cavities within the assembly that are less than one micrometer in size, there are not defined features of that scale.

Version 1:

Reviewer comments:

Referee #1

(Remarks to the Author)

I have examined the response and the revised manuscript. The authors have made an extensive effort in responding to reviewers, revising the paper and adding new material to the manuscript and the SI. The new tables in the SI would definitely present a valuable summary of the state of art in the field for the readers who read the paper to the extent of SI data. I can't see how this paper can be improved further beyond this point and would not object to its publication. However, there is one element where my comments still stand to some extent. I am still not sure that live cells would survive the optical assembly process, even if the laser is focused slightly away from the cell. Perhaps the authors could perform few simple experiments to check the viability of cells under these conditions, or at least be a bit less assertive in the concluding paragraph where they envision the method being useful for biomaterials, without providing a proof of that hypothesis.

Referee #2

(Remarks to the Author)

In my opinion, the authors have accurately addressed the comments raised by myself and the other reviewers. I appreciate the additional discussion on scope and limitations and the demonstration of extended nanofabrication capacities using very small particles of a wide range of materials. I remain of the opinion that the method is original, innovative and versatile with respect of materials and structures. As thus, I believe that it is of interest to the broadest community of scientist as targeted by Nature and recommend publication.

That said, I have a few remarks on the new data provided in the revised versions that I believe should be addressed by the reviewers:

New data in Figure 4. I appreciate the demonstration of the authors to pattern very small nanomaterials into defined shapes – as shown e.g. for titania and iron oxide. However, what I am missing is a higher resolution SEM image to complement the complete structure and the individual building blocks – so that the reader gets an impression on packing density, structure within the assembly, potential porosity, etc. The same is true for the cube-like structures, where EDX mapping show the different elements, but the different shapes of the particles that are "claimed" in the figure caption are not visible.

Updated Figure 5: The new device the authors demonstrate is neat as it can filter out even small particles. However, even more convincing would be a demonstration that some particles can also pass the assembly – i.e. as a sorting device instead of a filter. This should be easily feasible by adjusting the particle size and thus the porosity formed by the interstitial sites. Maybe the authors could comment on such possibilities or the limitations associated with this type of device.

Table S1: I appreciate the comparison of different types of assembly techniques and follow the argumentation of the authors about the advantages of their technique. I only politely disagree about the proposed resolution of the interfacial assembly, which the authors specify to $\sim 1\mu\text{m}$. Indeed, sub-micrometer particles are routinely assembled at liquid interfaces by many different research groups – see, e.g. (Chem. Rev. 2015, 115, 6265–6311), limited predominantly by the availability of monodisperse polystyrene or silica particles, which typically come in sizes $>100\text{nm}$. Even smaller nanocrystals can also be assembled at liquid interfaces – see, e.g. Chem. Rev. 2016, 116, 18, 11220–11289.

Response Letter

We thank the reviewers for their insightful and constructive feedback on our manuscript. We have modified and revised our manuscript based on the reviewer comments and suggestions, which have substantially improved the clarity of our work and highlighted its scientific contributions better. We provide point-by-point replies to the reviewer comments below, which are indicated by blue arrows (→) below. Additionally, we have listed all changes made in either the main text or Supplementary Information files of our manuscript and highlighted them in yellow.

Reviewer #1:

Summary

The manuscript presents results, analysis and demonstration of an “optofluidic” microfabrication method. Particles of various types are manipulated by using laser light and assemble inside templated shapes or within microfluidic devices. The work is of high scientific quality, and the manuscript includes excellent analysis of some of the key principles involved in the process based on colloidal stability theory. The results are very well illustrated, and the corresponding movies are impressive. Without doubt, this is a high-quality professional paper. However, it has some drawbacks when considered for a publication in a very high impact journal.

→ We sincerely thank the reviewer for the very positive evaluation of our work and for recognizing the scientific quality, analysis, and presentation of our manuscript. We greatly appreciate the encouraging comments regarding the novelty of the optofluidic microfabrication method. The constructive feedback we received has been instrumental in enhancing the overall quality and impact of our research. We have thoroughly revised our manuscript in response to these comments.

Comment #1:

The optofluidic method presented in the paper appears to be a modification of the optical thermophoresis patterning techniques that have attracted large attention recently and have been subjects of significant research, including in more mainstream publications. For an example of

recent perspective see: <https://pubs.acs.org/doi/10.1021/acs.langmuir.5c01023>. The current paper extends the physics and application area of this family of methods, but does not present in my view principally new advances that could not have been foreseen based on others' recent progress.

→ We thank the reviewer for this thoughtful comment and for pointing us to the recent perspective on optical thermophoresis patterning. After a thorough review of the literature, we agree that optical assembly techniques have been widely employed for patterning of colloidal particles, including optical tweezers, opto-thermophoretic assembly, opto-thermoelectric assembly, and related methods. Broadly, these techniques can be classified into two categories: 1) Techniques that leverage optical force to directly trap and manipulate particles (e.g., optical tweezers); 2) Techniques that utilize light-controlled fields (e.g., electric or temperature fields) to indirectly guide particle migration and assembly (e.g., thermophoretic or thermoelectric assembly). While existing techniques have enabled important progress, several limitations remain. For example:

1. Lack of generality across materials. Existing techniques do not provide a universal solution for a broad spectrum of materials. For instance, thermophoretic assembly is highly sensitive to experimental parameters, such as solvent composition, ambient temperatures, and particle surface properties, while optical tweezers are restricted to colloidal particles within a specific size range (typically ≈ 100 nm to 10 μ m).

2. Limited capability for freeform 3D architectures. Most reported assemblies are confined to 2D patterns, with only a few simple 3D structures demonstrated to date (e.g., out-of-plane 1D chain or microcube).

3. Low assembly efficiency. Existing optical assembly techniques operate in a one-by-one manner, resulting in limited throughput, typically on the order of 10^1 - 10^3 particles/min.

In contrast, the unique advantage of our technique lies in its ability to realize truly volumetric freeform 3D micro-/nanoarchitectures with broad material compatibility. We demonstrate its applicability to metal, metal oxides, diamond, and quantum dots, as well as nanowires with different aspect ratios (100-200) and micro-/nanoparticles across a wide size range (10 nm-10 μ m) and diverse surface properties. Our technique shows low sensitivity to solvent compositions and achieves a high assembly efficiency of up to 10^5 particles/min.

To provide a comprehensive comparison of the assembly capabilities between our technique and existing optical assembly techniques, we have revised the manuscript accordingly. In addition, we have included a detailed table in the Supplementary Information that summarizes and contrasts the features of our method with other representative assembly strategies, including optical assembly, capillary assembly, DNA-based assembly, and related techniques.

In the main text,

(Page 2, line 50) Among various assembly techniques, optical assembly²³⁻³⁰ has been an appealing strategy for the construction of complex material assemblies. It leverages the non-specific light-matter interactions (e.g., optical gradient forces³¹, light-controlled electric or temperature fields³²) to trap micro-/nanoparticles suspended in a solution. The trapped particles can then be transported and positioned one by one at designated locations, enabling high-precision single-particle assembly. Techniques such as optical tweezers³³, opto-thermophoretic assembly³⁴, opto-thermoelectric assembly³⁵, and optothermal flow-based assembly^{24,27,36,37} have been used to assemble colloidal particles with diverse sizes, shapes, and surface properties. However, these approaches remain largely limited to 2D structural configurations and exhibit low assembly efficiencies, typically on the order of 10^1 - 10^3 particles/min. More importantly, establishing a general optical assembly platform with broader material applicability is still challenging, as most techniques impose strict requirements on parameters, such as particle size/surface properties, solvent composition, and other environmental conditions. A detailed comparison of various assembly strategies is provided in Supplementary Table 1.

(Page 3, line 90) For example, our approach enables the fabrication of a series of solid microcubes randomly assembled by SiO₂ nanoparticles with an assembly efficiency of approximately 10^5 particles/min,

In Supplementary Materials,

Supplementary Table 1 Comparison among different assembly technologies.

Assembly techniques	Mechanism	Resolution	Material applicability	Structural complexity	Number of building blocks in assembled structures	Typical time scale or key parameters for assembly	Ref.
Capillary assembly	Capillary flows induced by the evaporation of colloidal suspensions transport dispersed particles toward the triple-phase contact line, guiding their assembly at predetermined substrate locations. 3D assembly configurations can be achieved with the assistance of a template.	Sub-1 μm	Accommodating a variety of micro-/nanoparticles, including polymer, metal, metal oxide, MOF, COF, etc.	2.5D or limited 3D geometries, e.g., cylinder and star-shaped Au NP superstructure	$\approx 10^2\text{-}10^4$	Upon evaporation, the receding speed of the triple-phase contact line is usually below 5 $\mu\text{m/s}$ to ensure effective particle filling of the template.	1,2
Interfacial assembly	Particles spontaneously adsorb at the liquid-liquid interface of two immiscible liquids (e.g., water and oil), where they assemble into organized interfacial patterns.	$\approx 1 \mu\text{m}$		2D lattice patterns or 3D structures are limited to spherical geometries.		Minutes	3,4
Electric field assembly	Charged metallic nanoparticles are directed onto the substrate using an electric field or a combination of electric and flow fields. By precisely moving the substrate in 3D space, complex 3D micro-/nanostructures can then be fabricated through the layer-by-layer assembly of particles.	Sub-100 nm	Metals, alloys	Free-standing complex 3D structures, e.g., helical or flower-shaped micro-/nanostructures.	/	To ensure the continuous growth of assembled structures, the moving speed of the substrate is usually below 100 nm s^{-1} .	5,6
Magnetic field assembly	Magnetic particles self-assemble into ordered superstructures through dipole-dipole interaction under an external magnetic field. By tuning the applied field, different assembly behaviours can be induced.		Magnetic particles	1D chain, 2D lattice, or limited 3D configurations, e.g., twisted ribbon.	$\approx 10^1\text{-}10^3$	Milliseconds to seconds	7-9
DNA-based assembly	DNA strands and DNA-functionalized colloidal particles self-assemble through the deterministic base-pairing interactions (A-T, C-G), analogous to the modular assembly of LEGO bricks.	Sub-10 nm (assembly of DNA molecules)	DNA molecules and DNA-functionalized colloidal particles	Complex DNA 3D nanostructures, such as monolith, railed bridge, and icosahedron. Colloidal superstructures with simple 3D configuration, for instance, analogous to AB-type molecules.	$\approx 10^1\text{-}10^2$	Hours to days	10-12
Polyphenol-based assembly	Polyphenol-functionalized particles first assemble through interfacial molecular interactions, and the resulting configurations are then stabilized via metal-ligand coordination.	$\approx 1 \mu\text{m}$	Polyphenol-functionalized colloidal particles	Limited 3D geometries, for instance, spherical and core-satellite configurations.	$\approx 10^2\text{-}10^3$	Seconds.	13

Optical assembly	Optical tweezer	An optical gradient force from a tightly focused laser beam is used to trap individual colloidal particles. By scanning the laser beam, the trapped particle can be transported and positioned at desired locations, enabling one-by-one assembly.	Single-particle resolution (Sub-1 μm)	Colloidal particles with appropriate size (e.g., 0.1-10 μm in diameter) ^{14,15} .	Limited 3D geometries, e.g., a cube-shaped superstructure.	$\approx 10^1$ - 10^3	The assembly efficiency is estimated to be around 10^0 - 10^1 particles/min.	16,17
	Thermophoretic assembly	Suspended particles in solution undergo directed thermophoretic migration toward either the hot or cold region when a temperature gradient is generated by laser irradiation. By tuning particle surface charge, solvent composition, and environmental temperature, particles can be trapped at the hot region (laser spot), enabling thermophoretic tweezer for assembly.		Colloidal particles with appropriate surface charges.	2D micro-/nanopatterns. 3D structures with fairly limited geometries, e.g., out-of-plane 1D chain made of 500 nm PS sphere ¹⁸ .		The migration speed of particles is normally below 100 $\mu\text{m/s}$. Otherwise, the drag force exerted by the fluid will release the trapped particle from the laser beam, hindering the assembly. The assembly efficiency is estimated to be around 10^1 - 10^2 particles/min.	18,19
	Thermoelectric assembly	In an electrolyte solution, various ions migrate at varying velocities and directions under a temperature gradient via thermophoresis. This imbalance generates a spatial distribution of ion species, giving rise to a thermoelectric field in the solution. The resulting electric field drives the directed migration of charged colloidal particles. By tuning experimental parameters, such as ion type and particle surface charge, colloidal particles can be selectively trapped at the laser spot, enabling an opto-thermoelectric tweezer for particle assembly.	Sub-1 μm	Broader material applicability irrespective of particle size, shape, or surface charge—for example, diverse micro-/nanoparticles and nanowires, including metal, metal oxide, carbon materials, and quantum dots.	2D micro-/nanostructures	$\approx 10^2$ - 10^4	The assembly efficiency is estimated to be around 10^2 - 10^3 particles/min.	20,21
	Convective flow-based assembly	A thermal gradient induces variations in density, pressure, and surface tension within the solution, generating strong convective flows. Suspended particles are carried along the streamlines, and ultimately assemble at the laser spot through van der Waals interactions.						
	Optofluidic 3D assembly (our work)	By harnessing strong convective flows induced by steep temperature gradients within a confined 3D space, various micro-/nanoparticles can be volumetrically assembled inside micro-confinements, yielding freeform 3D architectures entirely composed of micro-/nanoparticles.						

Comment #2:

The introduction of the manuscript presents the technique as an alternative to the common 2PP techniques. However, the implementation of the method to make the 3D structures presented here relies on first using 2PP for making the templates where the particles would be driven and assembled optofluidically.

→ We thank the reviewer for this insightful comment. We agree that the current implementation of our technique relies on 2PP to fabricate the microtemplates that subsequently guide the optofluidic assembly. We would like to clarify that our intention is not to position the method as a direct replacement of 2PP, but rather as a complementary approach that significantly extends the capabilities of conventional 2PP. While 2PP excels at fabricating polymeric micro-/nanostructures with high spatial resolution, it is intrinsically limited in material scope, being largely restricted to photosensitive resists. Our optofluidic method leverages 2PP-fabricated templates as guiding frameworks but enables the incorporation of a much broader range of functional materials—including metals, oxides, diamond, and quantum dots—into truly volumetric freeform 3D architectures. In this sense, 2PP provides the structural scaffold, while the optofluidic process provides the material versatility and assembly efficiency.

We emphasize that our technique represents an advanced strategy for 3D micro-/nanofabrication, as it not only overcomes the material limitations of conventional 2PP but also demonstrates superior capability in creating highly freeform, volumetric 3D micro-/nanostructures. To provide a comprehensive comparison with existing 3D micro-/nanofabrication techniques, we have added a detailed table in the Supplementary Materials and a graphical comparison in Supplementary Fig. 1. Furthermore, we have revised the Introduction section of the main text to more clearly highlight these distinctions.

In the main text,

(Page 2, line 74) By integrating the optofluidic assembly process with 2PP, our optofluidic 3D micro-/nanofabrication strategy proposes a generalizable paradigm that overcomes the material limitations of 2PP, opening new avenues for nanotechnology (Supplementary Fig. 1 and Table 2).

In Supplementary Materials,

Supplementary Fig. 1 Radar comparison of the common 2PP technique and our optofluidic 3D micro-/nanofabrication approach.

Supplementary Table 2 Comparison among different 3D micro-/nanofabrication techniques.

3D micro-/nanofabrication techniques	Mechanism	Resolution	Material applicability	Structural complexity	Ref.
Physical vapor deposition (PVD)	By evaporating a target source and condensing the vaporized species onto a substrate, 3D structures can be fabricated using glancing angle deposition (GLAD) through controlled “shadowing effect”.	Sub-100 nm	Metals, metal oxides, ceramics, semiconductors, and others.	Limited 3D geometries, e.g., pillars, helical structures, and nanotubes.	24,25
Lithographic methods (e.g., UV lithography, e-beam lithography)	A pattern is generated from a mask by selectively exposing a photoresist film, which is then transferred to the targeted substrate. 3D structures can be created either through repeated exposures or by inducing 3D photoresist exposure via spatially modulated light pattern.		Polymers, metals, metal oxides, semiconductors, and others.		26-28
Mechanically guided 3D assembly	2D structures can be transformed into 3D configurations through mechanical deformation process such as buckling, folding, and rolling.				29,30
Focused electron/ion beam-induced deposition (FEBID/FIBID)	Focused electron or ion beams locally decompose organometallic precursors, enabling the maskless direct writing of 3D nanostructures.	Sub-10 nm	Applicable materials are limited to metals, e.g., Au, Pt	Highly freeform 3D structures, e.g., tetragonal-bipyramid, icosahedron	31
Two-photon polymerization (2PP)	Nonlinear two-photon absorption (2PA) of a photoinitiator generates free radicals, triggering the chemical cross-linking of monomer molecules (photoresists) at the focal point of a tightly focused femtosecond laser beam. Scanning the laser beam in 3D space enables the direct laser writing (DLW) of 3D micro-/nanostructures.	Sub-1 μm	Printable materials can be expanded to metal, metal oxides, glass, quantum dots and more through the development of advanced photoresists with tailored chemistries. However, a general route applicable across diverse material systems remains challenging, and the mass loading of targeted materials in printed structures is typically limited to an upper bound of ≈90%.	Highly freeform 3D micro-/nanoarchitectures, e.g., 3D lattice microstructures.	32-43
	Nanomaterials can be deposited or adsorbed onto 2PP-printed polymer templates through chemical coating, PVD, or weak interaction-based absorption (e.g., electrostatic interaction, hydrogen bonding, and van der Waals force).		Broad material applicability, including metal, metal oxides, carbon materials, QDs, ceramics, and others. However, the resulting structures are not volumetric. Desired materials are merely deposited onto the template surface with low mass loadings (< 60%).		44
Optofluidic 3D micro-/nanofabrication (our work)	By harnessing the strong convective flow induced by fs laser heating within a confined 3D space, various micro-/nanoparticles can be volumetrically assembled inside the confinement, enabling the creation of freeform 3D architectures entirely composed of micro-/nanoparticles.		Broader material applicability irrespective of their size, shape, or surface charge—for example, diverse micro-/nanoparticles and nanowires, including metal, metal oxide, carbon materials, and quantum dots. The fabricated architectures are fully composed of the targeted materials, with mass loading approaching ≈100%.	Highly freeform 3D volumetric structures, e.g., a dangling croissant-shaped SiO ₂ superstructure.	

Comment #3:

The paper claims that the optofluidic technique can be applied to biological systems. However, the use of laser heating, even in scanning mode, is likely to lead to damage in delicate biomaterials and living systems. One element that would have provided useful information is the evaluation of the thermal impact and possible damage in biomolecular structures.

→We thank the reviewer for raising this question. To clarify, our claim is that the functional microfluidic **devices** fabricated by this technique can be applied to biological systems instead of being directly fabricated within biological systems. In these applications, there is no direct laser involved, and thus no risk of thermal damage to living systems. To avoid potential confusion, we have re-elaborated the related discussion in the conclusion section of the revised manuscript.

At the same time, the reviewer highlights an interesting direction for further expanding the material applicability of our optofluidic 3D micro-/nanofabrication technique to biomaterials and living systems. We noted that optical strategies have been widely explored to manipulate living cells, such as the opto-thermophoretic tweezers (*Lin, L et al. ACS Nano 2017, 11, 3147–3154*) and the convective flow-based assembly technique (*Makey G et al. Nature Physics, 2020, 16(7): 795-801*). Usually, the heating effect induced by a fs laser is largely confined to a narrow region around the laser spot, with limited influence on materials located farther away (*Ekici O et al. Journal of physics D: Applied physics, 2008, 41(18): 185501; Makey G et al. Nature Physics, 2020, 16(7): 795-801*). Importantly, by precisely controlling the flow field, our technique allows particle assembly within templates positioned away from the laser focus. This highlights the potential applicability for the targeted 3D assembly of biomaterials. We have added related discussions in the conclusion section of the revised manuscript.

In the main text,

(Page 8, line 294) In future research, we also envision extending this technique to biomaterials and living systems (e.g., cells), as the highly localized heating of the femtosecond laser is largely confined to focal region and is expected to cause minimal damage to materials positioned slightly away from the laser spot²⁷.

Comment #4:

The method is touted as an advancement in “nanofabrication”, yet it has been applied to particles that are closer to the microscale than the nanoscale. Some of the results, as the microsieveing in Fig. 5 and the materials scale shown in Figure 5g apply to the ability of any porous microstructure to capture objects in its matrix. Fig. 5g seems to imply that the method uses new principles of manipulating the objects shown, which is misleading.

→ We thank the reviewer for this constructive comment. We agree that many of the demonstrations in the current manuscript focus on the microscale, as our aim was to first establish the fundamental principles and showcase proof-of-concept applications. To further demonstrate the *nanofabrication* capability of our technique, we performed additional experiments to construct 3D architectures with nanoscale features using smaller nanoparticles, e.g., TiO₂ NPs (≈ 100 nm diameter) and Fe₃O₄ NPs (≈ 20 nm diameter). As shown in the updated Figs. 4m-r of the main text, we fabricated a TiO₂ NP-assembled screw-like microstructure with helical threads of ≈ 320 nm in width, as well as an Fe₃O₄ NP-assembled alphabet letter “E” with a height of ≈ 855 nm. We have included these data and corresponding discussions in the revised manuscript. Furthermore, to more accurately reflect the scope of our method, we have revised the title to “*Optofluidic three-dimensional micro- and nanofabrication*” and consistently replaced the term “*nanofabrication*” with “*micro-/nanofabrication*” throughout the manuscript.

In the main text:

(Page 6, line 235) Beyond these micrometer-sized SiO₂ particles, our method is compatible with a broad range of nanomaterials, enabling the fabrication of diverse 3D structures with nanoscale features. As illustrated in Figs. 4m-r, a screw-like microstructure with helical threads of around 320 nm in width (Figs. 4m-n) and an alphabet letter “E” with a height of around 855 nm (Figs. 4p-r) are created using TiO₂ NPs with 90.2 ± 15.8 nm diameter and Fe₃O₄ NPs with 16.7 ± 2.3 nm diameter, respectively.

In updated Figure 4,

Fig. 4 Wide compatibility with versatile micro-/nanomaterials. **a-c**, 3D model (**a**) and SEM images (**b**, **c**) of a micro-gourd superstructure assembled with 1 μm SiO_2 particles. **d-f**, 3D model (**d**) and SEM images (**e**, **f**) of a hexagon-shaped micro superstructure assembled with 600 nm SiO_2 particles. **g-i**, 3D model (**g**) and SEM images (**h**, **i**) of a microsphere co-assembled with 1 μm and 600 nm SiO_2 particles. **j-l**, Model (**j**) and SEM images (**k**, **l**) of alphabet letters superstructures “P” (assembled with 1 μm SiO_2 particles) and “T” (600 nm SiO_2 particles). **m**, **n**, 3D model (**m**) and SEM (**n**) image of a TiO_2 NP-assembled screw-like microstructure. **o**, Transmission electron microscopy (TEM) image of TiO_2 NPs. **p**, **q**, 3D model (**p**) and SEM images (**q-1**, **q-2**) of an alphabet letter “E” composed of Fe_3O_4 NPs. **r**, TEM image of Fe_3O_4 NPs. **s-z**, Model (**s**), SEM image (**t**), and energy-dispersive X-ray spectroscopy (EDS) mapping (**u-z**) of a microcube assembled with various materials, including SiO_2 (**t**), TiO_2 nanowires (NWs) (**u**), WO_3 NWs (**v**), diamond nanoparticles (NPs) (**w**), Al_2O_3 NWs (**x**), Fe_3O_4 NPs (**y**), and Ag NPs (**z**). Scale bars are 10 μm in (**b**, **e**, **h**, **k**, **t**), 4 μm in (**c**, **f**, **i**, **l**), 5 μm in (**u-z**), 2 μm in (**n**, **q-1**), 800 nm in **q-2**, 200 nm in **o**, and 80 nm in **r**.

Comment #5:

In summary, this is a paper that is rich in detail, demonstration of microfabricated structures and investigation and analysis of the basic colloidal stability phenomena involved in the optofluidic assembly process. The capabilities of the technique are demonstrated with numerous common types of particles and some of the structures and dynamics are quite

impressive visually. However, the underlying optothermal manipulation method has already been studied elsewhere and has inherent limitations. Thus, while this paper would make an excellent contribution to an above-average impact journal, I am not very impressed by its level of novelty and importance as compared to the extremely high expectations for publication in Nature.

→ We sincerely thank the reviewer for their careful evaluation and recognition of the strengths of our work, including the detailed analysis, comprehensive demonstrations, and the visually impressive structures and dynamics enabled by our technique. We respectfully acknowledge that our approach builds upon established physical principles. However, the novelty of our work lies in establishing a unified optofluidic framework for 3D micro-/nanofabrication that transcends prior optical assembly approaches, representing a paradigm shift by enabling volumetric freeform architectures, broad material generality, and device-level integration that were previously unattainable.

1) **Truly volumetric freeform 3D architectures at the micro-/nanoscales**, moving beyond the predominantly 2D and limited 3D demonstrations (e.g., out-of-plane 1D chain or microcube) in prior optical assembly work.

2) **Unprecedented material generality**, accommodating metals, oxides, diamond, quantum dots, nanowires, and nanoparticles across the 10 nm–10 μm range—surpassing the scope of conventional 2PP or single-physics optical methods.

3) **Scalable efficiency**, with deterministic assembly rates up to 10^5 particles per minute, orders of magnitude higher than one-by-one particle manipulation in other optical assembly approaches.

4) **Precise spatial control and multi-material integration**. Local fs-laser addressing enables distinct structures in proximity. This extends to functional devices, including a microfluidic chip with concatenated microvalves for selective nanoparticle sorting, and a multimodal microrobot integrating four materials exhibits three distinct motion modes under external stimuli (see in updated **Figure 5f-s**).

5) **New insights into colloidal 3D assemblies are enabled by a dynamic, flow-driven strategy** that overturns the conventional paradigm of static assembly—where external disturbances such as shaking or mixing must typically be avoided—thereby providing a new perspective for the field.

Together, these advances elevate our contribution beyond a methodological variation to a generalizable paradigm for fabricating complex, functional 3D architectures across diverse materials systems. We believe this step-change in fabrication capability—bridging optical physics, colloidal science, and device engineering—not only provides new insights into fundamental colloidal assembly but also opens new frontiers in fields such as reconfigurable photonics, multifunctional microdevices, and bio-integrated systems.

We have clarified this positioning more explicitly in the revised Introduction and Discussion to ensure that the broader conceptual and practical significance of our work is clear.

In updated Figure 5,

Fig. 5 On-demand construction of multifunctional microdevices. a-e, Microfluidic sieving devices. **a**, Schematic illustration showing the capillary-driven separation process of a microfluidic sieving device. **b**, Time-lapsed optical images showing the capillary-driven fluid flow through a microfluidic chip embedded with a colloidal microvalve (20 μm width, 40 μm length) composed of 1 μm SiO_2 particles. Images are extracted from Supplementary Video 21. **c**, SEM image of the SiO_2 -assembled microvalve. **d**, Fluorescent image of 500 nm polystyrene (PS) nanospheres rejected by the microvalve. **e**, Fluorescent image showing 100 nm PLGA

nanoparticles rejected by the microvalve. Inset is the variation of gray value along the microchannel. **f-s**, Multifield-driven microrobots. **f, g**, Schematic illustration (**f**) and trajectory (**g**) of the magnetic tumbling of a Fe_3O_4 NP-assembly cylinder microrobot. **h-o**, Schematic illustration (**h, j, l, m**) and motion trajectories (**i, k, m, o**) of TiO_2 -Au microrobots with different shapes and material distributions. **p-s**, Schematic illustration (**p**) of a L-shaped microrobot integrated with four different materials (TiO_2 , Au, Pt, and Fe_3O_4) and its three motion modes: magnetic pulling (**q**), light-driven counterclockwise rotation (**r**), and clockwise rotation in around 5 wt% H_2O_2 (**s**). Scale bars are 50 μm in (**b, d, e**), 60 μm in (**g, i, k, m, o, r, s**), 30 μm in **q**, and 10 μm in **c**.

Reviewer #2:

Summary

Creating free-form objects at smallest dimensions remains challenging. Despite intense developments in additive manufacturing techniques, the state-of-the-art processes (2-photon polymerization) are limited by a selected choice of (mostly polymeric) resins. Creating similar structures from functional materials may unlock a wide range of application in miniaturized devices, from chromatography and detection to nanophotonics and plasmonics. In their manuscript, Lyu et al. pick up this challenge and develop a bottom-up manufacturing process that delivers such functional, freeform 3D nanostructures from colloidal building blocks. The process uses polymeric 3D printed objects as templates and delivers particles within this cavity by optofluidic flow fields created by a femtosecond laser. They elucidate process conditions that trigger nucleation and cluster growth within these templates, until eventually, the object is completely filled and can be released by decomposing the template. The process described in this well-written and appealing manuscript is innovative, thoroughly characterized and the authors carefully elucidate the underlying physics. The process pushes the limits of nanofabrication and provides a new paradigm for the design of tailored functional, hybrid, and nanoporous materials at the smallest scale with a wide range of applications. In my view, the manuscript therefore appeals to a broadest community of readers at the intersection of science, technology, and manufacturing and is thus of interest for Nature. The following points may serve to further improve on the description and understanding of the manuscript:

→ We sincerely thank the reviewer for the positive and encouraging assessment of our work. We are very pleased that the reviewer finds the manuscript well written, appealing, and innovative, and that the process is recognized as pushing the limits of micro-/nanofabrication and providing a new paradigm for creating functional hybrid and nanoporous materials. We also greatly appreciate the reviewer's recognition of the broad potential applications of our technique and its relevance to a wide scientific and technological community. We have carefully considered the reviewer's constructive suggestions for further improving the clarity and understanding of the manuscript.

Comment #1:

Why is the templating form needed at all? It seems that for fundamental studies of the clustering threshold (e.g. Figure 2), the authors did not use any template at all and the process simply

operates by triggering clustering via overcoming the energy balance towards aggregation. I assume that the template helps keep a defined form (instead of a freeform “writing” process with less defined edges and symmetry), but this could be more clearly discussed in the manuscript.

→ We thank the reviewer for the constructive comment. Indeed, as the reviewer correctly points out, clustering and aggregation can be triggered without a template, as shown in our fundamental studies (e.g., Fig. 2). In these cases, the optofluidic flow field drives particles to nucleate and form clusters once the energy balance shifts towards aggregation. However, the template plays a crucial role in enabling deterministic 3D fabrication: it defines the overall geometry to ensure well-defined edges and symmetry, channels the optofluidic flow to reproducibly fill complex volumes, and provides design versatility by allowing diverse 3D architectures—such as croissant-shaped superstructures and nanoscale screw-like structures—that cannot be achieved through uncontrolled clustering.

To address the reviewer’s comment, we have added related discussions in the revised manuscript.

In the main text,

(Page 4, line 138) The template we used plays a crucial role in enabling deterministic 3D fabrication: it defines the overall geometry to ensure well-defined edges and symmetry, channels the optofluidic flow to reproducibly fill complex volumes, and provides design versatility by allowing diverse 3D architectures.

Comment #2:

When particles aggregate via vdW forces upon “delivery” via the optofluidic flow by overcoming electrostatic repulsion, why do they not directly clog the opening into the template? How can it be ensured that the entire hollow template fills up with particles efficiently (which seems to be the case in the objects the authors printed)?

→ We thank the reviewer for this insightful question. In our experiments, clogging at the template opening can sometime occur, particularly when interparticle interactions dominate over particle-fluid interactions. This effect is more pronounced for micrometer-sized particles, which have larger volumes and are hard to enter the cavity, and for nanoparticles near or below 100 nm, which experience strong adhesion, and tend to aggregate at the entrance. To illustrate this, we have added Supplementary Video 19, showing blocked openings with 3 μm and 200

nm SiO₂ particles in immersion oil. To overcome this issue and enable successful 3D assembly, we found that increasing the opening size or optimizing the hole configuration is effective. Additional experiments with optimized templates for assembling these materials (e.g., SiO₂ with a size range of 200 nm-10 μm, 20 nm Fe₃O₄) are included in the revised manuscript and Supplementary Information, together with corresponding discussions.

In Methods,

(Page 12, line 482) Finally, it should be noted that excessively strong interparticle attraction can lead to the formation of large clusters that clog the openings, preventing complete filling of the 3D structures (Supplementary Video 19). This issue can be mitigated by optimizing the size and distribution of template openings, as shown in Supplementary Note 1, Supplementary Figs. 18-20.

In Supplementary Materials,

Supplementary Fig. 18 Time-lapse optical images of the assembly process of 3 μm SiO₂ nanoparticles into hollow microcubes with varying opening sizes.

Supplementary Fig. 19 a, b, Critical opening sizes (a) for SiO₂ particles of different diameters during assembly in immersion oil and the corresponding 3D models (b).

Supplementary Fig. 20 Assembly of SiO₂ particles in a microcube template with multiple openings. **a**, 3D models of the microcube templates, featuring holes of two different sizes (8 μm and 3 μm) distributed on opposite sides of the cube. **b**, Schematic illustration of assembly process. When a laser beam is applied to the side with 3 μm holes (right), a directional fluid flow is generated from right to left, propelling dispersed SiO₂ particles through the 8 μm openings into the cube. The particles are then blocked by the smaller 3 μm holes, leading to continuous accumulation inside the cube until the template is fully filled.

Comment #3:

In the (co)assembly of binary mixtures, the authors demonstrate that they can either merge both populations or create pure structures from one and the other particle system, respectively. What determines which of the two cases occurs? I assume that different particle systems exhibit different tendencies to cluster (i.e. have a different clustering threshold). Is that correct and if so, there should be criteria that one object needs to be printed first (i.e. of the easier-to-cluster population). I little more detail here would be useful.

→ We thank the reviewer for raising this question. The outcome—whether co-assembly or the formation of distinct pure structures—depends primarily on the experimental protocol rather than on intrinsic clustering thresholds of the particles.

In the co-assembly experiment, a mixed suspension of two types of particles is employed. Both populations are simultaneously transported toward the template by the optofluidic flow and are jointly incorporated into the forming cluster, resulting in binary microstructures.

In the sequential pure-assembly experiments, a multi-step protocol is adopted. Suspensions of different particles are introduced successively, with a washing step in between to remove excess particles and prevent cross-interference. For instance, to fabricate the letters “P” and “I”

on a substrate, we first assembled “I” using a suspension of 600 nm SiO₂ particles. After rinsing with isopropanol and water, a suspension of 1 μm SiO₂ particles was then introduced to assemble “P.” By repeating these procedures, different particle systems can be used to construct distinct structures at predefined locations or to achieve seamless integration within a single microstructure.

To improve clarity, we have added more methodological details in the *Methods* section.

In Methods,

(Page 12, line 455) Additionally, co-assembly of different materials can be obtained by preparing a mixed suspension through physical blending of various particles, as demonstrated in Fig. 4i of the main text, where SiO₂ microspheres with 600 nm and 1 μm diameter were co-assembled into a single structure.

(Page 12, line 472) To create 3D structures or devices made from different particles, we employed a sequential, multi-step assembly procedure. Suspensions of different particles were introduced successively, with a washing step between each stage to remove excess particles and prevent cross-interference. For instance, when creating alphabet letters “P” and “I”, a suspension of 600 nm SiO₂ was first used to assemble “I”. The excess suspension was then rinsed away with IPA and water to ensure clean conditions for the next step, after which a suspension of 1 μm SiO₂ particles was introduced to assemble “P”. By repeating this processes, multiple particle systems can be used to construct distinct structures at predefined locations on a single substrate or to achieve seamless integration within a single microstructure, thereby enabling the creation of multifunctional microdevices.

Comment #4:

Why do the particles seemingly assemble in a well-ordered form (or is that impression of the images, e.g. Fig 1 misleading), if they are produced by a diffusion-limited aggregation process? In my view, to create ordered structures, the particles need to be able to probe the energy landscape at least to a certain extend (i.e. need to have sufficient time to assemble). Maybe the authors could comment on this aspect a bit.

→ We thank the reviewer for pointing out this issue. In our technique, the particles are assembled into 3D superstructures through an optofluidic, diffusion-limited aggregation process, and thus the resulting assemblies are not well-ordered lattice structures (Figure 1b, c). The previous schematic figures in Figs 1a, g, and Fig. 2c may cause some misinterpretation.

Accordingly, we have revised these schematics in the updated manuscript to depict particles in a more realistic, slightly random configuration. We have also added clarifying text in the main Discussion to emphasize that the assemblies are stochastic superstructures rather than perfectly ordered lattices.

In the main text,

(Page 3, line 90) For example, our approach enables the fabrication of a series of solid microcubes randomly assembled by SiO₂ nanoparticles with an assembly efficiency of approximately 10⁵ particles/min, as shown in the scanning electron microscope (SEM) image in Fig. 1b, its enlarged image in Fig. 1c, and Supplementary Fig. 2.

In revised Figure 1,

Fig. 1 Concept of optofluidic 3D micro-/nanofabrication. **a**, Schematic illustration of the optofluidic 3D micro-/nanofabrication process, where a localized temperature gradient induced by a femtosecond (fs) laser heating, generates a strong convective flow, guiding the 3D assembly of micro-/nanoparticles within a confined hollow 3D microtemplate, printed by two-photon polymerization. **b**, **c**, Scanning electron microscopy (SEM) images of a SiO₂ colloidal particle-assembled microcube (**b**) and its zoomed-in view (**c**). **d**, **e**, SEM images of a dangling

croissant-shaped microstructure with 3D curved surface assembled from SiO₂ particles (**d**) and its enlarged view (**e**). The inset in (**d**) is the 3D model of the croissant structure. **f**, Simulation result showing the temperature distribution and fluid flow field around a hollow microcube upon fs laser heating. **g**, Schematic illustration and time-lapse optical images showing the assembly process of SiO₂ nanoparticles within a hollow microcube. These images are extracted from Supplementary Video 1.

In revised Figure 2c,

Fig. 2c, Schematic illustration and optical images depicting the assembly process of SiO₂ particles within a hollow microcube in 1M NaCl solution.

Comment #5:

It would add to the manuscript if the authors discuss some boundary conditions of the process. These could include: i) the critical size of the hole (maybe with respect to the entire template, or the particle size used to assemble the freeform); ii) the sizes of the particles that can be used (can they be too small or too big, and if so, why); iii) the laser energy (can the bubble from overheating be too large)?; and iv) the obtainable size of the structures (will crack formation occur upon a critical size)? This would be useful for a reader to fairly assess how to apply the process to his/her needs and requirements.

→ We thank the reviewer's constructive suggestions, and have accordingly conducted additional experiments to provide more detailed information about our technique.

i: The critical size of the hole. As discussed in comment #2, the template opening can be clogged either by very small nanoparticles or larger micrometer-sized particles. To prevent such clogging, a sufficiently large opening is required, and thus the critical hole size highly depends on particle diameter. To clarify this, we conducted additional experiments to determine the critical hole sizes for SiO₂ particles of different sizes. The result are now summarized in the Supplementary Fig. 19 in the revised Supplementary Materials.

Supplementary Fig. 19 a, b, Critical opening sizes (a) for SiO₂ particles of different diameters during assembly in immersion oil and the corresponding 3D models (b).

ii: The size of the particles that can be used. By optimizing the design of the template, we have successfully assembled various micro-/nanoparticles, with a wide range of 10 nm-10 μm. The detailed information has been summarized in the updated Supplementary Fig. 11, Supplementary Video 11, and Supplementary Table 3.

Supplementary Fig. 11. Morphology of various nanomaterials used in this work. a, SEM image of a microcube assembled with TiO₂ NWs. Inset is the SEM image of TiO₂ NWs. b, SEM image of WO₃ NWs. c-f, TEM images of diamond NPs (c), Al₂O₃ NWs (d), Au NPs (e), and Ag NPs (f).

Supplementary Table 3. The physicochemical properties of micro-/nanomaterials used in this work.

Materials	Size	Concentration used for assembly	Solvent used for assembly
SiO ₂	150 nm - 10 μm	1 wt% for 150 nm SiO ₂ , and 0.5 wt% for others	Immersion oil, mineral oil, oleic acid, or 1 mM CTAB
WO ₃ NWs	50 nm × 10 μm	around 2 mg/mL	Mineral oil
Ag NPs	< 100 nm		Mineral oil
Al ₂ O ₃ NWs	2-6 nm × 200-400 nm		Mineral oil
Fe ₃ O ₄ NPs	≈ 200 nm		Mineral oil
TiO ₂ NWs	100 nm × 10 μm		Mineral oil
diamond	≈ 10 nm		Immersion oil
CdTe Quantum dots	/		1 mM CTAB
Au NPs	≈ 50 nm		50 mM CTAB
Pt NPs	≈ 50 nm		50 mM CTAB
TiO ₂ NPs	≈ 100 nm		DMSO
Fe ₃ O ₄ NPs	≈ 20 nm		10 mM CTAB
SiO ₂ mixtures	600 nm, 1 μm	1 wt%	Immersion oil

iii) The laser energy. Adjusting the laser power and scan speed can lead to varying laser dosages. A low laser dosage results in a weak flow field. When the flow field is too weak, particles cannot be efficiently driven into the cavity of the template, resulting in the failure to assemble into 3D structures. To illustrate this, we have included updated Supplementary Video 5, which shows that particles fail to fill the entire template in immersion oil under a weak flow field with a flow speed of approximately 40 μm/s. This flow field is generated at a low laser dosage with a laser power of 20 mW and a scan speed of 500 μm/s.

Regarding bubble formation, we observed a large bubble (≈100 μm in diameter) at a laser power of 50 mW (scan speed: 300 μm/s). Importantly, this bubble had no observable influence on particle assembly within the template in immersion oil. We have revised our manuscript to clarify the importance of the flow field and laser energy.

In the main text,

(Page 3, line 107) Additionally, we noticed that bubble formation with a diameter up to 100 μm (Supplementary Fig. 4) occurs easily due to solvent evaporation induced by femtosecond

laser heating, leading to an additional Marangoni flow³⁹, and this disturbance further promotes and accelerates particle assembly within the template (Supplementary Video 1).

(Page 5, line 173) Besides, an extremely weak flow field cannot provide sufficient driving force to continuously transport the particles toward the inside of a template, failing to fill up the template (Supplementary Video 5).

In Supplementary Materials,

Supplementary Fig. 4 Optical image of bubble formation during the assembly of SiO₂ particles in immersion oil. The bubble appears at a laser power of ≈ 50 mW (maximum output of the instrument) with a scanning speed of $300 \mu\text{m/s}$.

Supplementary Fig. 7. Assembly speed of SiO₂ particles as a function of the laser power (red spheres) and scan speed (blue squares). The assembly experiments were conducted in immersion oil, with SiO₂ particle dispersions at a concentration of ≈ 0.5 wt%. The laser power was set to 50 mW for the scan speed series. The scan speed was set at 500 $\mu\text{m/s}$ for the laser power series. It is also noted that a lower laser power of 20 mW (with a scan speed of 500 $\mu\text{m/s}$) would lead to an extremely weak flow field with a flow speed of around 40 $\mu\text{m/s}$, which cannot provide sufficient driving force to ensure particles fill up the entire template (Supplementary Video 5).

iv: Obtainable size of the structures. It is easy to fabricate microstructures with a feature size below 100 μm . For instance, we fabricated a series of SiO₂ particle-assembled microcubes with a size range of 5-40 μm . They all show good structural integrity. To create structures larger than 100 μm , however, a much stronger flow field than we used throughout this work is expected to be required to provide sufficient driving force for continuously transporting particles toward the inside of the template, ensuring the hollow template can be entirely filled with particles. We have included the discussions and corresponding results in the revised manuscript and Supplementary Materials.

In the main text,

(Page 3, line 90) For example, our approach enables the fabrication of a series of solid microcubes randomly assembled by SiO₂ nanoparticles with an assembly efficiency of approximately 10^5 particles/min, as shown in the scanning electron microscope (SEM) image in Fig. 1b, its enlarged image in Fig. 1c, and Supplementary Fig. 2.

In Supplementary Materials,

Supplementary Fig. 2 SEM images of SiO₂ colloidal-assembled microcubes of different sizes. The white layer on the particle surface is Au sputtered for SEM characterization. Microstructures with feature sizes below 100 μm can be readily fabricated. However, for structures larger than 100 μm, a much stronger flow field than that employed in this work is required to provide sufficient driving force for continuous particle transport into the template and to ensure complete filling of the hollow cavity.

Comment #6:

The description of the role of surfactants seems a bit oversimplified. On page 6, the authors mention that “surfactants introduce [...] enhanced DLVO force”, further promoting colloidal aggregation”. The behavior very much depends on the nature of the surfactant. In their case, a negatively-charged silica particle will indeed be destabilized by the addition of CTAB, but not because of it being a surfactant per se, but because of the positive charge. SDS for example would likely promote stability and thus hinder the process.

→ We thank the reviewer for this constructive comment. We agree with the reviewer that the effect of surfactants on colloidal stability strongly depends on their chemical nature and interaction with the particle surface, rather than on their general classification as “surfactants”. In our case, the negatively charged silica particles are indeed destabilized by the addition of CTAB, primarily due to the introduction of positive charges that neutralize or reverse the surface charge, thereby lowering the electrostatic repulsion barrier and promoting aggregation. By contrast, an anionic surfactant such as SDS would likely enhance stability by further increasing the negative surface charge and thus hinder the assembly process. However, we also observed the cluster formation in SDS, which might be contributed to other interactions involved in this process, such as thermophoretic force and depletion force. We have included a detailed discussion and an updated Supplementary Fig. 8 elucidating the role of different surfactants in the clustering process.

In the main text,

(Page 6, line 202) Moreover, the cationic ions (CTA^+) dissociated from CTAB molecules absorb onto the surface of negatively charged SiO_2 particles, neutralizing their surface charge (Supplementary Fig. 8). This reduction in electrostatic repulsion enhances the DLVO attractions among the SiO_2 particles, thereby promoting their aggregation.

(Page 6, line 211) Notably, SiO_2 particles exhibit strong electrostatic repulsion in SDS solution (Supplementary Fig. 8), which may hinder cluster formation. However, other interparticle attractions, such as opto-thermophoretic force^{23,34,47} and depletion force⁴⁸, may also arise in these surfactant systems, facilitating particle aggregation.

In Supplementary Materials,

Supplementary Fig. 8. Zeta potential of SiO_2 colloidal particles (1 μm in diameter) in various solutions.

Comment #7:

On P7 and in Figure 4, the authors demonstrate the use of a range of particle systems. In some cases, the order and homogeneity of the print seems to be markedly worse (e.g. Fe_3O_4 or Ag; Figure 4 q, w). Why is that?

→ We thank the reviewer for raising this question. The poor order and homogeneity of the assemblies are strongly influenced by the uniformity of particle size distribution. The Fe_3O_4

NPs used in the previous Figure 4q have a broad size distribution of 194.9 ± 103.2 nm, which resulted in poor structural homogeneity. To address this, we performed additional experiments using Fe_3O_4 nanoparticles with a much narrower size distribution (16.7 ± 2.26 nm), which yielded high-quality microstructures with smoother surfaces, as shown in the updated Figs. 4p–r in the main text. These new results and corresponding discussions have been included in the revised manuscript and Supplementary Materials.

In the main text,

(Page 7, line 241) Notably, smaller nanoparticles with a uniform in size distribution yield micro-/nanostructures with much smoother surfaces (Fig. 4q, Supplementary Fig. 10).

In updated Fig. 4,

Fig. 4 Wide compatibility with versatile micro-/nanomaterials. **a-c**, 3D model (**a**) and SEM images (**b**, **c**) of a micro-gourd superstructure assembled with $1 \mu\text{m}$ SiO_2 particles. **d-f**, 3D model (**d**) and SEM images (**e**, **f**) of a hexagon-shaped micro superstructure assembled with 600 nm SiO_2 particles. **g-i**, 3D model (**g**) and SEM images (**h**, **i**) of a microsphere co-assembled with $1 \mu\text{m}$ and 600 nm SiO_2 particles. **j-l**, Model (**j**) and SEM images (**k**, **l**) of alphabet letters superstructures “P” (assembled with $1 \mu\text{m}$ SiO_2 particles) and “I” (600 nm SiO_2 particles). **m**, **n**, 3D model (**m**) and SEM (**n**) image of a TiO_2 NP-assembled screw-like microstructure. **o**, Transmission electron microscopy (TEM) image of TiO_2 NPs. **p**, **q**, 3D model (**p**) and SEM images (**q-1**, **q-2**) of an alphabet letter “E” composed of Fe_3O_4 NPs. **r**, TEM image of Fe_3O_4 NPs. **s-z**, Model (**s**), SEM image (**t**), and energy-dispersive X-ray spectroscopy (EDS) mapping

(u-z) of a microcube assembled with various materials, including SiO₂ (t), TiO₂ nanowires (NWs) (u), WO₃ NWs (v), diamond nanoparticles (NPs) (w), Al₂O₃ NWs (x), Fe₃O₄ NPs (y), and Ag NPs (z). Scale bars are 10 μm in (b, e, h, k, t), 4 μm in (c, f, i, l), 5 μm in (u-z), 2 μm in (n, q-1), 800 nm in q-2, 200 nm in o, and 80 nm in r.

In Supplementary Materials,

Supplementary Fig. 10 Surface morphology of microstructures assembled from Fe₃O₄ NPs with different size distribution. a, SEM image (left) of the zoomed-in surface of the alphabet letter “E” assembled from Fe₃O₄ NPs with a uniform size distribution (16.7 ± 2.3 nm), and the TEM image (right) of the corresponding NPs. **b**, SEM image (left) of the zoomed-in surface of a microcube assembled from Fe₃O₄ NPs with a broad size distribution (194.9 ± 103.2 nm), and the SEM image (right) of the corresponding NPs.

Comment #8:

For the case of Ag nanoparticles, are localized surface plasmon resonances excited with the laser light NPs and does the resulting plasmonic heating interfere with the process (or can it be used to make the process more efficient)?

→ We thank the reviewer for raising this question. In our experiments, the fs laser is primarily couples to the liquid medium rather than directly to the nanoparticles, since the particle volume fraction is quite low. Thus, the dominant mechanism driving assembly is thermally induced

fluid convection arising from localized laser heating of the solvent. While Ag nanoparticles can indeed support localized surface plasmon resonances under near-infrared excitation, their contribution is secondary in our system. Specifically, plasmonic heating may occur when Ag nanoparticles accumulate at the laser focus, which can further amplify local temperature gradients and strengthen convective flows. So this effect would act synergistically with the general optofluidic mechanism.

Comment #9:

How are hydrophilic SiO₂ particles dispersed in oil? In my experience, such systems will not be colloiddally stable, but remain agglomerated. Did the authors characterize the size-distribution?

→ We thank the reviewer for raising this question. We agreed with the reviewer that the hydrophilic SiO₂ particles will become unstable when dispersed in oil. We have conducted additional experiments characterizing the stability of SiO₂ particles in immersion oil. The result shows that they can remain dispersed for at least 1h, which is much longer than the time we typically need (several minutes) for assembling experiments. This is due to the high viscosity of the oil (≈ 500 mPs·s), which highly limits the Brownian motion of SiO₂ particles, preventing their agglomeration within a short period of time. We have accordingly revised our manuscript and Supplementary Materials.

In Methods,

(Page 12, line 450) These hydrophilic SiO₂ particles can remain dispersed in various oils within the experimental timescale due to the high viscosity of the medium, which helps prevent clogging the template openings by large agglomerations during assembly (Supplementary Fig. 17).

In Supplementary Materials,

Supplementary Fig. 17. Time-lapse optical images showing that SiO₂ particles remain dispersed in immersion oil for at least 1 h. The highly viscous solvent (≈ 500 mPs·s) suppresses Brownian motion, thereby preventing particle agglomeration.

Comment #10:

In the experimental section, the authors mention that they apply oxygen plasma for 6h. Is that really needed and would the subsequent annealing step not suffice to consolidate the structures?

→ We thank the reviewer for raising this question. Plasma etching is indeed an efficient method for removing polymers under mild conditions. In our process, applying O₂ plasma prior to the annealing step helps maintain structural integrity during template removal. As shown in Fig. R1, we used a plasma cleaner with relatively low working power, which requires extended processing times to completely remove the template. For higher efficiency, we also employ a plasma asher with stronger power (Fig. R2), enabling template removal within a few dozen minutes and thereby improving overall fabrication efficiency. To clarify the role of plasma etching in our process, we have included a detailed discussion in Methods in the main text.

[FIGURE REDACTED]

Fig. R1. **[TEXT REDACTED]**

[FIGURE REDACTED]

Fig. R2. [TEXT REDACTED]

In Methods,

(Page 13, line 487) To completely remove the outer-layer polymer template and maintain the structural integrity, the colloidal-assembled microstructures were subjected to mild O₂ plasma treatment for 6 h with a plasma cleaner machine (Tergeo-EM, PIE Scientific) at 75 W power and an O₂ flow rate of 20 sccm. To further enhance interfacial bonding, the microstructures were annealed in air at 600°C for 2 h with a heating rate of 2°C/min. For faster template removal, a high-power Plasma Asher (MPR-6) can also be employed.

Reviewer #3:

Summary

In this paper the authors present a 3D Optofluidic “Nanofabrication” technique and demonstrate the assembly of different shapes, an assembled filtration system, and perform an analysis on the mechanism. The technique relies on creating a “template” of sorts using two photon polymerization and then filling the cavity with beads using optically driven thermal convective flow. As noted, a variety of shapes are created and a fairly substantive analysis is performed to understand some of the basic physics behind the process. Ultimately though I think the contribution here fall below the standard of what I would expect for a publication in Nature – primarily on the basis of novelty. A few reasons

→ We sincerely thank the reviewer for careful evaluation and recognizing the breadth of the demonstrations and substantive mechanistic analysis in our manuscript. The constructive feedback we received has played a pivotal role in elevating the overall quality and impact of our research. In response to the insightful comments from the reviewer, we have conducted additional experiments and analyses to further strengthen both the novelty and significance of our study. We believe that our work now constitutes an important and valuable contribution to the field of study. To better highlight the innovation inherent in our approach, we have summarized the innovative aspects into the following key points (1-6):

1) **Truly volumetric freeform 3D architectures at the micro-/nanoscales**, moving beyond the predominantly 2D and limited 3D demonstrations (e.g., out-of-plane 1D chain or microcube) in prior optical assembly work. Examples include the dangling croissant-shaped superstructure with intricate 3D curved surfaces shown in Figs. 1d, e, the screw-like microstructure with nanoscale helical threads in the updated Figs. 4m-n, and the alphabet letter “E” with a height of approximately 855 nm in the updated Figs. 4q-r.

2) **Unprecedented material generality**, accommodating metals, oxides, diamond, quantum dots, nanowires, and nanoparticles across the 10 nm–10 μm range—surpassing the scope of conventional 2PP or single-physics optical methods. Conventional 2PP is intrinsically restricted to a narrow set of cross-linkable polymers. Besides, existing optical assembly techniques do not provide a universal solution for a broad spectrum of materials due to their strict criteria on parameters, such as the particle size/surface properties and solvent composition. In contrast, our work offers **a general solution that leverages non-specific optofluidic**

interactions to transport and guide diverse micro-/nanoparticles, assembling them into a 2PP-printed hollow microtemplate, **enabling the creation of freeform 3D micro-/nanoarchitectures with a broad spectrum of materials.**

3) **Scalable efficiency**, with deterministic assembly rates up to 10^5 particles per minute, several orders of magnitude higher than one-by-one particle manipulation in other optical assembly approaches (10^1 - 10^3 particles per minute). In most optical assembly techniques, individual particles are first trapped at the focal point of a laser, then transported and placed at the desired position on a substrate, achieving the on-demand assembly. The assembly efficiency is highly restricted to the migration speed of particles, which is usually slower than $100\ \mu\text{m/s}$, leading to a slow assembly rate. In contrast, our technique leverages a strong convective flow (several mm/s) to transport suspended particles into a hollow microtemplate, enabling the fast assembly of up to 10^5 particles/min.

4) **Precise spatial control and multi-material integration.** Local fs-laser addressing enables distinct structures in close proximity without cross-interference. This capability extends to functional devices, including a microfluidic chip with concatenated microvalves for selective nanoparticle sorting, and a multifunctional microrobot integrating four materials that exhibits three distinct motion modes under external stimuli.

5) **New insights on colloidal assemblies via a dynamic, flow-driven route**, beyond the limitations of conventional colloidal assembly under static conditions. Usually, aggregation in colloidal systems is assumed to occur under static conditions, where any external disturbance—such as shaking or mixing—is deliberately avoided to prevent disrupting the natural balance of interparticle interactions. In contrast, particles undergo rapid coagulation by increasing the nanoparticle velocity in our method. The increased velocity of particles enhances the frequency and energy of collisions, enabling particles to overcome electrostatic repulsion and promote particle aggregation. These advances establish our assembly strategy as a new paradigm that provides fresh insights into the fundamental and applied aspects of colloidal assembly.

Together, these advances elevate our contribution beyond a methodological variation to a generalizable paradigm for fabricating complex, functional 3D architectures across diverse materials systems. We believe this step-change in fabrication capability—bridging optical physics, colloidal science, and device engineering—will open up new frontiers in fields such as reconfigurable photonics, multifunctional microdevices, and bio-integrated systems.

We have clarified this positioning more explicitly in the revised Introduction and Discussion to ensure that the broader conceptual and practical significance of our work is clear. Moreover, we have conducted additional experiments and discussions to emphasize these key novelty points. These include **the experiments on the fabrication of 3D structures with nanoscale features, demonstrations of creating multifield-driven microrobots with multimodal locomotion, as well as the detailed comparison between existing 3D micro-/nanofabrication techniques and our newly developed technique.** We extend our gratitude to the reviewer for their valuable insights, which have greatly improved our work. We hope that the reviewer finds the revised version of our manuscript to be an improved and informative contribution to the field.

In the main text,

(Page 2, line 42) Recent efforts have aimed to expand printable materials beyond polymers, primarily through developing advanced photoresists with tailored chemistries, **such as grafting photochemically bonding ligands onto inorganic colloidal nanocrystals^{14,15} or incorporating metal-coordination complexes into cross-linkable monomers¹⁶⁻²².** Nonetheless, these approaches remain restricted to specific materials and continue to face challenges in achieving broad compatibility with diverse material systems.

(Page 2, line 50) **Among various assembly techniques, optical assembly²³⁻³⁰ has been an appealing strategy for the construction of complex material assemblies. It leverages the non-specific light-matter interactions (e.g., optical gradient forces³¹, light-controlled electric or temperature fields³²) to trap micro-/nanoparticles suspended in a solution. The trapped particles can then be transported and positioned one by one at designated locations, enabling high-precision single-particle assembly. Techniques such as optical tweezers³³, opto-thermophoretic assembly³⁴, opto-thermoelectric assembly³⁵, and optothermal flow-based assembly^{24,27,36,37} have been used to assemble colloidal particles with diverse sizes, shapes, and surface properties. However, these approaches remain largely limited to 2D structural configurations and exhibit low assembly efficiencies, typically on the order of 10^1 - 10^3 particles/min. More importantly, establishing a general optical assembly platform with broader material applicability is still challenging, as most techniques impose strict requirements on parameters, such as particle size/surface properties, solvent composition, and other environmental conditions. A detailed comparison of various assembly strategies is provided in Supplementary Table 1.**

(Page 2, line 74) By integrating the optofluidic assembly process with 2PP, our optofluidic 3D micro-/nanofabrication strategy proposes a generalizable paradigm that overcomes the material limitations of 2PP, opening new avenues for nanotechnology (Supplementary Fig. 1 and Table 2).

(Page 3, line 90) For example, our approach enables the fabrication of a series of solid microcubes randomly assembled by SiO₂ nanoparticles with an assembly efficiency of approximately 10⁵ particles/min, as shown in the scanning electron microscope (SEM) image in Fig. 1b, its enlarged image in Fig. 1c, and Supplementary Fig. 2.

(Page 6, line 235) Beyond these micrometer-sized SiO₂ particles, our method is compatible with a broad range of nanomaterials, enabling the fabrication of diverse 3D structures with nanoscale features. As illustrated in Figs. 4m-r, a screw-like microstructure with helical threads of around 320 nm in width (Figs. 4m-n) and an alphabet letter “E” with a height of around 855 nm (Figs. 4p-r) are created using TiO₂ NPs with 90.2±15.8 nm diameter and Fe₃O₄ NPs with 16.7±2.3 nm diameter, respectively.

(Page 7, line 252) With the aid of precise spatial control and broad material applicability, microstructures spatially encoded with diverse functional materials can be created, highlighting the potential of our technique for developing microdevices with on-demand functionalities.

(Page 7, line 273) Subsequently, we demonstrate the application of our technique in fabricating multifield-driven microrobots with multimodal locomotion (Figs. 4f-s). As illustrated in Figs. 4f, g, a Fe₃O₄ NP-assembled cylinder microrobot exhibits magnetically controlled tumbling on the substrate (Supplementary Video 13). The magnetic response can be flexibly tuned by regulating the Fe₃O₄ NP content (Supplementary Fig. 15). Beyond magnetic actuation, Figs. 4h-o illustrate light-driven microrobots with controlled motion enabled by tailoring both the shape and spatial distribution of functional materials. For examples, a cylinder heterojunction TiO₂-Au microrobot demonstrates linear propulsion via self-electrophoresis⁴⁹ under UV illumination in water (Figs. 4h, i, Supplementary Fig. 16, and Supplementary Video 14). Shape asymmetry further enables rotational motion: a T-shaped TiO₂-Au microrobot rotates in tight circles (Figs. 4j, k, Supplementary Video 15) while a V-shaped design produces larger circular trajectories (Figs. 4l, m, Supplementary Video 16). Moreover, alternative spatial encoding of functional materials allows the same V-shaped microrobot to switch from rotation to linear motion (Figs. 4n, o, Supplementary Video 17). Importantly, our technique also enables the integration of multiple functional materials into a single microrobot, achieving multi-stimulus

responsiveness. For instance, we fabricate an L-shaped microrobot incorporating Au, TiO₂, Pt, and Fe₃O₄ (Figs. 4p-s), which exhibits three distinct motion modes: magnetic pulling, counterclockwise rotation under UV light, and clockwise rotation in H₂O₂ (Supplementary Video 18).

(Page 8, line 297) We believe this step-change in fabrication capability—bridging optical physics, colloidal science, and device engineering—not only provides new insights into fundamental colloidal assembly but also opens new frontiers in various fields, such as reconfigurable photonics, multifunctional microdevices/robots, and bio-integrated systems.

In updated Figure 4,

Fig. 4 Wide compatibility with versatile micro-/nanomaterials. **a-c**, 3D model (**a**) and SEM images (**b**, **c**) of a micro-gourd superstructure assembled with 1 μm SiO₂ particles. **d-f**, 3D model (**d**) and SEM images (**e**, **f**) of a hexagon-shaped micro superstructure assembled with 600 nm SiO₂ particles. **g-i**, 3D model (**g**) and SEM images (**h**, **i**) of a microsphere co-assembled with 1 μm and 600 nm SiO₂ particles. **j-l**, Model (**j**) and SEM images (**k**, **l**) of alphabet letters superstructures “P” (assembled with 1 μm SiO₂ particles) and “I” (600 nm SiO₂ particles). **m**, **n**, 3D model (**m**) and SEM (**n**) image of a TiO₂ NP-assembled screw-like microstructure. **o**, Transmission electron microscopy (TEM) image of TiO₂ NPs. **p**, **q**, 3D model (**p**) and SEM images (**q-1**, **q-2**) of an alphabet letter “E” composed of Fe₃O₄ NPs. **r**, TEM image of Fe₃O₄ NPs. **s-z**, Model (**s**), SEM image (**t**), and energy-dispersive X-ray spectroscopy (EDS) mapping

(u-z) of a microcube assembled with various materials, including SiO₂ (t), TiO₂ nanowires (NWs) (u), WO₃ NWs (v), diamond nanoparticles (NPs) (w), Al₂O₃ NWs (x), Fe₃O₄ NPs (y), and Ag NPs (z). Scale bars are 10 μm in (b, e, h, k, t), 4 μm in (c, f, i, l), 5 μm in (u-z), 2 μm in (n, q-1), 800 nm in q-2, 200 nm in o, and 80 nm in r.

In updated Figure 5,

Fig. 5 On-demand construction of multifunctional microdevices. a-e, Microfluidic sieving devices. a, Schematic illustration showing the capillary-driven separation process of a microfluidic sieving device. b, Time-lapsed optical images showing the capillary-driven fluid flow through a microfluidic chip embedded with a colloidal microvalve (20 μm width, 40 μm length) composed of 1 μm SiO₂ particles. Images are extracted from Supplementary Video 21. c, SEM image of the SiO₂-assembled microvalve. d, Fluorescent image of 500 nm polystyrene (PS) nanospheres rejected by the microvalve. e, Fluorescent image showing 100 nm PLGA nanoparticles rejected by the microvalve. Inset is the variation of gray value along the microchannel. f-s, Multifield-driven microrobots. f, g, Schematic illustration (f) and trajectory (g) of the magnetic tumbling of a Fe₃O₄ NP-assembly cylinder microrobot. h-o, Schematic illustration (h, j, l, m) and motion trajectories (i, k, m, o) of TiO₂-Au microrobots with different shapes and material distributions. p-s, Schematic illustration (p) of a L-shaped microrobot

integrated with four different materials (TiO₂, Au, Pt, and Fe₃O₄) and its three motion modes: magnetic pulling (**q**), light-driven counterclockwise rotation (**r**), and clockwise rotation in around 5 wt% H₂O₂ (**s**). Scale bars are 50 μm in (**b**, **d**, **e**), 60 μm in (**g**, **i**, **k**, **m**, **o**, **r**, **s**), 30 μm in **q**, and 10 μm in **c**.

In Supplementary Materials,

Supplementary Fig. 1 Radar comparison of the common 2PP technique and our optofluidic 3D micro-/nanofabrication approach.

Supplementary Fig. 15. Averaged speeds (left) and schematic (right) of cylinder microrobots assembled with various contents of Fe_3O_4 NP. The magnetic strength is 10 mT.

Supplementary Table 1 Comparison among different assembly technologies.

Assembly techniques	Mechanism	Resolution	Material applicability	Structural complexity	Number of building blocks in assembled structures	Typical time scale or key parameters for assembly	Ref.
Capillary assembly	Capillary flows induced by the evaporation of colloidal suspensions transport dispersed particles toward the triple-phase contact line, guiding their assembly at predetermined substrate locations. 3D assembly configurations can be achieved with the assistance of a template.	Sub-1 μm	Accommodating a variety of micro-/nanoparticles, including polymer, metal, metal oxide, MOF, COF, etc.	2.5D or limited 3D geometries, e.g., cylinder and star-shaped Au NP superstructure	$\approx 10^2\text{-}10^4$	Upon evaporation, the receding speed of the triple-phase contact line is usually below 5 $\mu\text{m/s}$ to ensure effective particle filling of the template.	1,2
Interfacial assembly	Particles spontaneously adsorb at the liquid-liquid interface of two immiscible liquids (e.g., water and oil), where they assemble into organized interfacial patterns.	$\approx 1 \mu\text{m}$		2D lattice patterns or 3D structures are limited to spherical geometries.		Minutes	3,4
Electric field assembly	Charged metallic nanoparticles are directed onto the substrate using an electric field or a combination of electric and flow fields. By precisely moving the substrate in 3D space, complex 3D micro-/nanostructures can then be fabricated through the layer-by-layer assembly of particles.	Sub-100 nm	Metals, alloys	Free-standing complex 3D structures, e.g., helical or flower-shaped micro-/nanostructures.	/	To ensure the continuous growth of assembled structures, the moving speed of the substrate is usually below 100 nm s^{-1} .	5,6
Magnetic field assembly	Magnetic particles self-assemble into ordered superstructures through dipole-dipole interaction under an external magnetic field. By tuning the applied field, different assembly behaviours can be induced.		Magnetic particles	1D chain, 2D lattice, or limited 3D configurations, e.g., twisted ribbon.	$\approx 10^1\text{-}10^3$	Milliseconds to seconds	7-9
DNA-based assembly	DNA strands and DNA-functionalized colloidal particles self-assemble through the deterministic base-pairing interactions (A-T, C-G), analogous to the modular assembly of LEGO bricks.	Sub-10 nm (assembly of DNA molecules)	DNA molecules and DNA-functionalized colloidal particles	Complex DNA 3D nanostructures, such as monolith, railed bridge, and icosahedron. Colloidal superstructures with simple 3D configuration, for instance, analogous to AB-type molecules.	$\approx 10^1\text{-}10^2$	Hours to days	10-12
Polyphenol-based assembly	Polyphenol-functionalized particles first assemble through interfacial molecular interactions, and the resulting configurations are then stabilized via metal-ligand coordination.	$\approx 1 \mu\text{m}$	Polyphenol-functionalized colloidal particles	Limited 3D geometries, for instance, spherical and core-satellite configurations.	$\approx 10^2\text{-}10^3$	Seconds.	13

Optical assembly	Optical tweezer	An optical gradient force from a tightly focused laser beam is used to trap individual colloidal particles. By scanning the laser beam, the trapped particle can be transported and positioned at desired locations, enabling one-by-one assembly.	Single-particle resolution (Sub-1 μm)	Colloidal particles with appropriate size (e.g., 0.1-10 μm in diameter) ^{14,15} .	Limited 3D geometries, e.g., a cube-shaped superstructure.	$\approx 10^1$ - 10^3	The assembly efficiency is estimated to be around 10^0 - 10^1 particles/min.	16,17
	Thermophoretic assembly	Suspended particles in solution undergo directed thermophoretic migration toward either the hot or cold region when a temperature gradient is generated by laser irradiation. By tuning particle surface charge, solvent composition, and environmental temperature, particles can be trapped at the hot region (laser spot), enabling thermophoretic tweezer for assembly.		Colloidal particles with appropriate surface charges.	2D micro-/nanopatterns. 3D structures with fairly limited geometries, e.g., out-of-plane 1D chain made of 500 nm PS sphere ¹⁸ .		The migration speed of particles is normally below 100 $\mu\text{m/s}$. Otherwise, the drag force exerted by the fluid will release the trapped particle from the laser beam, hindering the assembly. The assembly efficiency is estimated to be around 10^1 - 10^2 particles/min.	18,19
	Thermoelectric assembly	In an electrolyte solution, various ions migrate at varying velocities and directions under a temperature gradient via thermophoresis. This imbalance generates a spatial distribution of ion species, giving rise to a thermoelectric field in the solution. The resulting electric field drives the directed migration of charged colloidal particles. By tuning experimental parameters, such as ion type and particle surface charge, colloidal particles can be selectively trapped at the laser spot, enabling an opto-thermoelectric tweezer for particle assembly.	Sub-1 μm	Broader material applicability irrespective of particle size, shape, or surface charge—for example, diverse micro-/nanoparticles and nanowires, including metal, metal oxide, carbon materials, and quantum dots.	2D micro-/nanostructures	$\approx 10^2$ - 10^4	The assembly efficiency is estimated to be around 10^2 - 10^3 particles/min.	20,21
	Convective flow-based assembly	A thermal gradient induces variations in density, pressure, and surface tension within the solution, generating strong convective flows. Suspended particles are carried along the streamlines, and ultimately assemble at the laser spot through van der Waals interactions.						
	Optofluidic 3D assembly (our work)	By harnessing strong convective flows induced by steep temperature gradients within a confined 3D space, various micro-/nanoparticles can be volumetrically assembled inside micro-confinements, yielding freeform 3D architectures entirely composed of micro-/nanoparticles.						
				Highly freeform 3D architectures, e.g., a dangling croissant-shaped superstructure	$\approx 10^2$ - 10^5	Assembly can occur at flow speeds of up to several mm/s with an estimated assembly efficiency of up to $\approx 10^5$ particles/min.		

Supplementary Table 2 Comparison among different 3D micro-/nanofabrication techniques.

3D micro-/nanofabrication techniques	Mechanism	Resolution	Material applicability	Structural complexity	Ref.
Physical vapor deposition (PVD)	By evaporating a target source and condensing the vaporized species onto a substrate, 3D structures can be fabricated using glancing angle deposition (GLAD) through controlled “shadowing effect”.	Sub-100 nm	Metals, metal oxides, ceramics, semiconductors, and others.	Limited 3D geometries, e.g., pillars, helical structures, and nanotubes.	24,25
Lithographic methods (e.g., UV lithography, e-beam lithography)	A pattern is generated from a mask by selectively exposing a photoresist film, which is then transferred to the targeted substrate. 3D structures can be created either through repeated exposures or by inducing 3D photoresist exposure via spatially modulated light pattern.		Polymers, metals, metal oxides, semiconductors, and others.		26-28
Mechanically guided 3D assembly	2D structures can be transformed into 3D configurations through mechanical deformation process such as buckling, folding, and rolling.				29,30
Focused electron/ion beam-induced deposition (FEBID/FIBID)	Focused electron or ion beams locally decompose organometallic precursors, enabling the maskless direct writing of 3D nanostructures.	Sub-10 nm	Applicable materials are limited to metals, e.g., Au, Pt	Highly freeform 3D structures, e.g., tetragonal-bipyramid, icosahedron	31
Two-photon polymerization (2PP)	Nonlinear two-photon absorption (2PA) of a photoinitiator generates free radicals, triggering the chemical cross-linking of monomer molecules (photoresists) at the focal point of a tightly focused femtosecond laser beam. Scanning the laser beam in 3D space enables the direct laser writing (DLW) of 3D micro-/nanostructures.	Sub-1 μm	Printable materials can be expanded to metal, metal oxides, glass, quantum dots and more through the development of advanced photoresists with tailored chemistries. However, a general route applicable across diverse material systems remains challenging, and the mass loading of targeted materials in printed structures is typically limited to an upper bound of ≈90%.	Highly freeform 3D micro-/nanoarchitectures, e.g., 3D lattice microstructures.	32-43
	Nanomaterials can be deposited or adsorbed onto 2PP-printed polymer templates through chemical coating, PVD, or weak interaction-based absorption (e.g., electrostatic interaction, hydrogen bonding, and van der Waals force).		Broad material applicability, including metal, metal oxides, carbon materials, QDs, ceramics, and others. However, the resulting structures are not volumetric. Desired materials are merely deposited onto the template surface with low mass loadings (< 60%).		44
Optofluidic 3D micro-/nanofabrication (our work)	By harnessing the strong convective flow induced by fs laser heating within a confined 3D space, various micro-/nanoparticles can be volumetrically assembled inside the confinement, enabling the creation of freeform 3D architectures entirely composed of micro-/nanoparticles.		Broader material applicability irrespective of their size, shape, or surface charge—for example, diverse micro-/nanoparticles and nanowires, including metal, metal oxide, carbon materials, and quantum dots. The fabricated architectures are fully composed of the targeted materials, with mass loading approaching ≈100%.	Highly freeform 3D volumetric structures, e.g., a dangling croissant-shaped SiO ₂ superstructure.	

Comment #1:

Optically driven thermal flows and particle concentration has been demonstrated many times in the past and similar, perhaps more extensive, analysis has been performed.

→We thank the reviewer for raising this question. We agree that optical assembly techniques have been widely employed for patterning of colloidal particles, including optical tweezers, opto-thermophoretic assembly, opto-thermoelectric assembly, and optothermal flow-based assembly. Broadly, these techniques can be classified into two categories: 1) Techniques that leverage optical force to directly trap and manipulate particles (e.g., optical tweezers); 2) Techniques that utilize light-controlled fields (e.g., electric or temperature fields) to indirectly guide particle migration and assembly (e.g., thermophoretic or thermoelectric assembly). While existing techniques have enabled important progress, several limitations remain. For example:

1. Lack of generality across materials. Existing techniques do not provide a universal solution for a broad spectrum of materials. For instance, thermophoretic assembly is highly sensitive to experimental parameters, such as solvent composition, ambient temperatures, and particle surface properties, while optical tweezers are restricted to colloidal particles within a specific size range (typically ≈ 100 nm to 10 μ m).

2. Limited capability for freeform 3D architectures. Most reported assemblies are confined to 2D patterns, with only a few simple 3D structures demonstrated to date (e.g., out-of-plane 1D chain or microcube).

3. Low assembly efficiency. Existing optical assembly techniques operate in a one-by-one manner, resulting in limited throughput, typically on the order of 10^1 - 10^3 particles/min.

In contrast, the unique advantage of our technique lies in its ability to realize truly volumetric freeform 3D micro-/nanoarchitectures with broad material compatibility. We demonstrate its applicability to metal, metal oxides, diamond, and quantum dots, as well as nanowires with different aspect ratios (100-200) and micro-/nanoparticles across a wide size range (10 nm-10 μ m) and diverse surface properties. Our technique shows low sensitivity to solvent compositions and achieves a high assembly efficiency of up to 10^5 particles/min. Additionally, from a fundamental perspective, our work opens a new insight of designing arbitrary colloidal assemblies via a dynamic, flow-driven route, beyond the limitations of conventional assembly under static conditions.

We have accordingly revised the Introduction and main text to clarify the advantages of our technique. Additionally, we have included a detailed table in the Supplementary Materials that summarizes and contrasts the features of our method with other representative assembly strategies, including optical assembly, capillary assembly, DNA-based assembly, and related techniques. For a more detailed examination of these revisions, please refer to our comprehensive response in the "Reply to the **Summary**" section.

(Page 2, line 50) Among various assembly techniques, optical assembly²³⁻³⁰ has been an appealing strategy for the construction of complex material assemblies. It leverages the non-specific light-matter interactions (e.g., optical gradient forces³¹, light-controlled electric or temperature fields³²) to trap micro-/nanoparticles suspended in a solution. The trapped particles can then be transported and positioned one by one at designated locations, enabling high-precision single-particle assembly. Techniques such as optical tweezers³³, opto-thermophoretic assembly³⁴, opto-thermoelectric assembly³⁵, and optothermal flow-based assembly^{24,27,36,37} have been used to assemble colloidal particles with diverse sizes, shapes, and surface properties. However, these approaches remain largely limited to 2D structural configurations and exhibit low assembly efficiencies, typically on the order of 10^1 - 10^3 particles/min. More importantly, establishing a general optical assembly platform with broader material applicability is still challenging, as most techniques impose strict requirements, on parameters such as particle size/surface properties, solvent composition, and other environmental conditions. A detailed comparison of various assembly strategies is provided in Supplementary Table 1.

Comment #2:

Ultimately this is just using that technique to fill up a cavity that is created by 2 photon polymerization. Given that this is required (or a similar technique) to create truly interesting shapes – its unclear to me why you couldn't just create the shapes using 2PP to begin with. Arguments in the paper are not convincing.

→We thank the reviewer for raising this question. It is correct that our current implementation relies on two-photon polymerization (2PP) or a similar templating technique to define the initial cavity. However, we emphasize that our approach is not simply “filling up” a 2PP cavity, but rather integrating 2PP with an optofluidic assembly process that enables a range of advances not achievable by 2PP alone. First, 2PP is intrinsically restricted to a narrow set of photosensitive polymeric resists, whereas our method allows these polymeric templates to be deterministically filled with a broad spectrum of functional materials (metals, oxides, diamond,

quantum dots, nanowires, nanoparticles). This overcomes the principal material limitation of 2PP. Second, the optofluidic process enables **continuous, high-efficiency transport and assembly** of particles into complex 3D volumes, reaching throughput ($\approx 10^5$ particles/min) far beyond what is achievable in one-by-one optical manipulation methods. Third, by tuning the balance of interparticle and particle–fluid interactions, our approach provides a **generalizable physical mechanism** that can be extended beyond polymer templates, offering design flexibility for future fabrication platforms. Thus, while 2PP provides the structural scaffolding, it is the optofluidic assembly that unlocks material diversity, scalability, and device-level functionality. We have revised the Introduction and Discussion to clarify this complementary relationship, positioning our method not as a substitute for 2PP but as a paradigm that **extends its utility into domains inaccessible to 2PP alone**. Additionally, we have included a detailed table and a graphical comparison in Supplementary Materials that summarizes and contrasts the features of our method with other representative 3D micro-/nanofabrication strategies. Please refer to our comprehensive response in the "Reply to the **Summary**" section for a more detailed examination of these revisions.

In the main text,

(Page 2, line 42) Recent efforts have aimed to expand printable materials beyond polymers, primarily through developing advanced photoresists with tailored chemistries, **such as grafting photochemically bonding ligands onto inorganic colloidal nanocrystals^{14,15} or incorporating metal-coordination complexes into cross-linkable monomers¹⁶⁻²²**. Nonetheless, these approaches remain restricted to specific materials and continue to face challenges in achieving broad compatibility with diverse material systems.

(Page 2, line 74) **By integrating the optofluidic assembly process with 2PP, our optofluidic 3D micro-/nanofabrication strategy proposes a generalizable paradigm that overcomes the material limitations of 2PP, opening new avenues for nanotechnology (Supplementary Fig. 1 and Table 2).**

Comment #3:

The demonstrated application of assembly of particle filters within a microfluidic chip has been demonstrated previously and the application itself is not very interesting. They discuss filtration of viruses and much smaller particles but only show a very simple demonstration.

→ We thank the reviewer for this constructive comment. We agree with the reviewer that microfluidic chips for the separation of micro-/nano-objects have been demonstrated previously. In our manuscript, this example was intended as a proof-of-concept to illustrate device-level integration. According to your comment, we have conducted additional experiments on fabricating multifunctional microrobots with multimodal locomotion, to further underscore the versatility and impact of our technique.

Microrobots are powerful devices that can perform complex tasks at the micro-/nanoscale, showing great promise across fields, such as environmental monitoring, targeted drug delivery, and minimally invasive surgery. Multifunctional integration can significantly expand microrobots' capabilities, applicability, and adaptability. Leveraging the precise spatial control and the broad material applicability of our technique, we created multifield-powered microrobots with controllable movements via spatially encoding diverse functional materials. As examples, we created microrobots powered by a magnetic field and light, respectively. These microrobots showcase controllable movements, including linear and rotational motion. Furthermore, a L-shaped microrobot spatially integrated with four distinct materials, including Au, TiO₂, Pt, and Fe₃O₄, is also demonstrated. This microrobot shows three different motion modes in response to different external stimuli. We believe these on-demand constructed microrobots would not only advance practical applications in biomedicine and environmental remediation, but also promote the fundamental research in the field of active matter as a popular model system.

To better demonstrate the versatility of our technique, we have included these data and discussions in the revised manuscript and Supplementary Materials.

In the main text,

(Page 7, line 273) Subsequently, we demonstrate the application of our technique in fabricating multifield-driven microrobots with multimodal locomotion (Figs. 4f-s). As illustrated in Figs. 4f, g, a Fe₃O₄ NP-assembled cylinder microrobot exhibits magnetically controlled tumbling on the substrate (Supplementary Video 13). The magnetic response can be flexibly tuned by regulating the Fe₃O₄ NP content (Supplementary Fig. 15). Beyond magnetic actuation, Figs. 4h-o illustrate light-driven microrobots with controlled motion enabled by tailoring both the shape and spatial distribution of functional materials. For examples, a cylinder heterojunction TiO₂-Au microrobot demonstrates linear propulsion via self-electrophoresis⁴⁹ under UV illumination in water (Figs. 4h, i, Supplementary Fig. 16, and Supplementary Video 14). Shape

asymmetry further enables rotational motion: a T-shaped TiO_2 -Au microrobot rotates in tight circles (Figs. 4j, k, Supplementary Video 15) while a V-shaped design produces larger circular trajectories (Figs. 4l, m, Supplementary Video 16). Moreover, alternative spatial encoding of functional materials allows the same V-shaped microrobot to switch from rotation to linear motion (Figs. 4n, o, Supplementary Video 17). Importantly, our technique also enables the integration of multiple functional materials into a single microrobot, achieving multi-stimulus responsiveness. For instance, we fabricate an L-shaped microrobot incorporating Au, TiO_2 , Pt, and Fe_3O_4 (Figs. 4p-s), which exhibits three distinct motion modes: magnetic pulling, counterclockwise rotation under UV light, and clockwise rotation in H_2O_2 (Supplementary Video 18).

In revised Figure 5,

Fig. 5 On-demand construction of multifunctional microdevices. a-e, Microfluidic sieving devices. **a**, Schematic illustration showing the capillary-driven separation process of a microfluidic sieving device. **b**, Time-lapsed optical images showing the capillary-driven fluid flow through a microfluidic chip embedded with a colloidal microvalve ($20\ \mu\text{m}$ width, $40\ \mu\text{m}$ length) composed of $1\ \mu\text{m}$ SiO_2 particles. Images are extracted from Supplementary Video 21.

c, SEM image of the SiO₂-assembled microvalve. **d**, Fluorescent image of 500 nm polystyrene (PS) nanospheres rejected by the microvalve. **e**, Fluorescent image showing 100 nm PLGA nanoparticles rejected by the microvalve. Inset is the variation of gray value along the microchannel. **f-s**, Multifield-driven microrobots. **f, g**, Schematic illustration (**f**) and trajectory (**g**) of the magnetic tumbling of a Fe₃O₄ NP-assembly cylinder microrobot. **h-o**, Schematic illustration (**h, j, l, m**) and motion trajectories (**i, k, m, o**) of TiO₂-Au microrobots with different shapes and material distributions. **p-s**, Schematic illustration (**p**) of a L-shaped microrobot integrated with four different materials (TiO₂, Au, Pt, and Fe₃O₄) and its three motion modes: magnetic pulling (**q**), light-driven counterclockwise rotation (**r**), and clockwise rotation in around 5 wt% H₂O₂ (**s**). Scale bars are 50 μm in (**b, d, e**), 60 μm in (**g, i, k, m, o, r, s**), 30 μm in **q**, and 10 μm in **c**.

In Supplementary Materials,

Supplementary Fig. 15. Averaged speeds (left) and schematic (right) of cylinder microrobots assembled with various contents of Fe₃O₄ NP. The magnetic strength is 10 mT.

Supplementary Fig. 16. Optical image of a TiO₂-Au heterojunction within a cylinder microtemplate.

Comment #4:

Finally this is a very generous use of the term nanofabrication. While there may be cavities within the assembly that are less than one micrometer in size, there are not defined features of that scale.

→ We thank the reviewer for this constructive comment. We agree that many of the demonstrations in the current manuscript focus on the microscale, as our aim was to first establish the fundamental principles and showcase proof-of-concept applications. To further demonstrate the *nanofabrication* capability of our technique, we performed additional experiments to construct 3D architectures with nanoscale features using smaller nanoparticles, e.g., TiO₂ NPs (≈100 nm diameter) and Fe₃O₄ NPs (≈20 nm diameter). As shown in the updated Figs. 4m-r of the main text, we fabricated a TiO₂ NP-assembled screw-like microstructure with helical threads of ~320 nm in width, as well as an Fe₃O₄ NP-assembled alphabet letter “E” with a height of ~855 nm. We have included these data and corresponding discussions in the revised manuscript. Furthermore, to more accurately reflect the scope of our method, we have revised the title to “*Optofluidic three-dimensional micro- and nanofabrication*” and consistently replaced the term “*nanofabrication*” with “*micro-/nanofabrication*” throughout the manuscript.

In the main text:

(Page 6, line 235) Beyond these micrometer-sized SiO₂ particles, our method is compatible with a broad range of nanomaterials, enabling the fabrication of diverse 3D structures with nanoscale features. As illustrated in Figs. 4m-r, a screw-like microstructure with helical threads of around 320 nm in width (Figs. 4m-n) and an alphabet letter “E” with a height of around 855 nm (Figs. 4p-r) are created using TiO₂ NPs with 90.2 ± 15.8 nm diameter and Fe₃O₄ NPs with 16.7 ± 2.3 nm diameter, respectively.

In updated Figure 4,

Fig. 4 Wide compatibility with versatile micro-/nanomaterials. **a-c**, 3D model (**a**) and SEM images (**b**, **c**) of a micro-gourd superstructure assembled with 1 μm SiO_2 particles. **d-f**, 3D model (**d**) and SEM images (**e**, **f**) of a hexagon-shaped micro superstructure assembled with 600 nm SiO_2 particles. **g-i**, 3D model (**g**) and SEM images (**h**, **i**) of a microsphere co-assembled with 1 μm and 600 nm SiO_2 particles. **j-l**, Model (**j**) and SEM images (**k**, **l**) of alphabet letters superstructures “P” (assembled with 1 μm SiO_2 particles) and “I” (600 nm SiO_2 particles). **m**, **n**, 3D model (**m**) and SEM (**n**) image of a TiO_2 NP-assembled screw-like microstructure. **o**, Transmission electron microscopy (TEM) image of TiO_2 NPs. **p**, **q**, 3D model (**p**) and SEM images (**q-1**, **q-2**) of an alphabet letter “E” composed of Fe_3O_4 NPs. **r**, TEM image of Fe_3O_4 NPs. **s-z**, Model (**s**), SEM image (**t**), and energy-dispersive X-ray spectroscopy (EDS) mapping (**u-z**) of a microcube assembled with various materials, including SiO_2 (**t**), TiO_2 nanowires (NWs) (**u**), WO_3 NWs (**v**), diamond nanoparticles (NPs) (**w**), Al_2O_3 NWs (**x**), Fe_3O_4 NPs (**y**), and Ag NPs (**z**). Scale bars are 10 μm in (**b**, **e**, **h**, **k**, **t**), 4 μm in (**c**, **f**, **i**, **l**), 5 μm in (**u-z**), 2 μm in (**n**, **q-1**), 800 nm in **q-2**, 200 nm in **o**, and 80 nm in **r**.

Response Letter

We appreciate the reviewers for their positive feedback on our manuscript. We have modified and revised our manuscript based on the reviewer comments and suggestions. We provide point-by-point replies to the reviewer comments below, which are indicated by blue arrows (→) below. Additionally, we have listed all changes made in either the main text or Supplementary Information files of our manuscript and highlighted them in yellow.

Reviewer #1:

Summary

I have examined the response and the revised manuscript. The authors have made an extensive effort in responding to reviewers, revising the paper and adding new material to the manuscript and the SI. The new tables in the SI would definitely present a valuable summary of the state of art in the field for the readers who read the paper to the extent of SI data. I can't see how this paper can be improved further beyond this point and would not object to its publication.

→ We sincerely thank the reviewer for the positive evaluation of our revised manuscript.

Comment #1:

However, there is one element where my comments still stand to some extent. I am still not sure that live cells would survive the optical assembly process, even if the laser is focused slightly away from the cell. Perhaps the authors could perform few simple experiments to check the viability of cells under these conditions, or at least be a bit less assertive in the concluding paragraph where they envision the method being useful for biomaterials, without providing a proof of that hypothesis.

→ We thank the reviewer for this valuable suggestion. We agree that our outlook regarding potential applications to living cells was not sufficiently supported by experimental data at this stage. To maintain a rigorous and evidence-based conclusion, we have removed this statement from the revised manuscript.

Reviewer #2:

Summary

In my opinion, the authors have accurately addressed the comments raised by myself and the other reviewers. I appreciate the additional discussion on scope and limitations and the demonstration of extended nanofabrication capacities using very small particles of a wide range of materials. I remain of the opinion that the method is original, innovative and versatile with respect of materials and structures. As thus, I believe that it is of interest to the broadest community of scientist as targeted by Nature and recommend publication. That said, I have a few remarks on the new data provided in the revised versions that I believe should be addressed by the reviewers:

→ We sincerely thank the reviewer for the positive evaluation of our work. We have revised our manuscript in response to the reviewer's valuable suggestions for further improving the clarity and understanding of the manuscript.

Comment #1:

New data in Figure 4. I appreciate the demonstration of the authors to pattern very small nanomaterials into defined shapes – as shown e.g. for titania and iron oxide. However, what I am missing is a higher resolution SEM image to complement the complete structure and the individual building blocks – so that the reader gets an impression on packing density, structure within the assembly, potential porosity, etc. The same is true for the cube-like structures, where EDX mapping show the different elements, but the different shapes of the particles that are “claimed” in the figure caption are not visible.

→ We thank the reviewer for the valuable suggestion. We agree that high-resolution SEM images showing the detailed surface morphology of assembled structures are helpful for catching the important information, such as packing density and porosity, and we thus updated Extended Data Fig. 1, which shows high-resolution surface morphology of microstructures assembled with TiO₂ NPs and Fe₃O₄ NPs with different sizes and size distributions. We also provide the SEM and TEM images of nanoparticles in Extended Data Figs. 1, 2. Accordingly, we made the following changes in the revised manuscript.

In the updated Extended Data Fig. 1,

b. SEM image (left) of the zoomed-in surface of a cylinder assembled from TiO₂ NPs with a size distribution of 90.2± 15.8 nm, and the TEM image (right) of the corresponding NPs.

In the caption of Fig. 4 (Line 471, page 21),

The high-resolution surface morphology of the microstructures assembled with different nanomaterials, along with the corresponding component nanomaterials, can be found in Extended Data Figs.1, 2.

Comment #2:

Updated Figure 5: The new device the authors demonstrate is neat as it can filter out even small particles. However, even more convincing would be a demonstration that some particles can also pass the assembly – i.e. as a sorting device instead of a filter. This should be easily feasible by adjusting the particle size and thus the porosity formed by the interstitial sites. Maybe the authors could comment on such possibilities or the limitations associated with this type of device.

→ We thank the reviewer for this suggestion. In fact, we have previously demonstrated this sorting capability in Supplementary Fig. 12. To make it clearer, a detailed description of the experimental results has been added to the caption of the Supplementary Fig. 12.

In the caption of Supplementary Fig. 12

This microfluidic chip enables the sequential separation of mixed micro-/nanoparticles, e.g., 2.5 μm and 500 nm PS particles. The 500 nm PS particles successfully pass through the first

valve (5 μm SiO_2 assembly) but are retained by the second microvalve (1 μm SiO_2 assembly).
In contrast, 2.5 μm PS spheres are rejected by the first valve (5 μm SiO_2 assembly).

Comment #3:

Table S1: I appreciate the comparison of different types of assembly techniques and follow the argumentation of the authors about the advantages of their technique. I only politely disagree about the proposed resolution of the interfacial assembly, which the authors specify to $\sim 1\mu\text{m}$. Indeed, sub-micrometer particles are routinely assembled at liquid interfaces by many different research groups – see, e.g. (Chem. Rev. 2015, 115, 6265–6311), limited predominantly by the availability of monodisperse polystyrene or silica particles, which typically come in sizes $>100\text{nm}$. Even smaller nanocrystals can also be assembled at liquid interfaces – see, e.g. Chem. Rev. 2016, 116, 18, 11220–11289.

→ We thank the reviewer for pointing out this question. We agree that the resolution of interfacial assembly can be up to several nanometers. We have updated the resolution of interfacial assembly into “sub-10 nm” in Table S1 and also cited the related literature.